# Direct nanopatterning of complex 3D surfaces and self-aligned superlattices via molecular-beam holographic lithography

Shuangshuang Zeng ®[1,2,4], Tian Tian[3,4], Jiwoo Oh ®[2], Zhan-Hong Lin ®[2] & Chih-Jen Shih ®[2] ✉

Conventional lithography methods involving pattern transfer through resist templating face challenges of material compatibility with various process solvents. Other approaches of direct material writing often compromise pattern complexity and overlay accuracy. Here we explore a concept based on the Moiré interference of molecular beams to directly pattern complex three-dimensional (3D) surfaces made by any evaporable materials, such as metals, oxides and organic semiconductors. Our proposed approach, termed the molecular-beam holographic lithography (MBHL), relies on precise control over angular projections of material flux passing through nanoapertures superimposed on the substrate, emulating the interference of coherent laser beams in interference lithography. Incorporating with our computational lithography (CL) algorithm, we have demonstrated self-aligned overlay of multiple material patterns to yield binary up to quinary superlattices, with a critical dimension and overlay accuracy on the order of 50 and 2 nm, respectively. The process is expected to substantially expand the boundary of materials combination for high-throughput fabrication of complex super-structures of translational symmetry on arbitrary substrates, enabling emerging nanoimaging, sensing, catalysis, and optoelectronic devices.

The development of materials nanotechnology has been largely driven by the evolution of nanopatterning techniques, giving rise to the fabrication of intricate 3D nanoscale structures for electronics[1], optics[2], bioengineering[3] and sensing[4] applications. The state-of-the-art nanopatterning techniques can be categorized into two major groups: the top-down and bottom-up approaches. The former is based on nanolithography transferring patterns through polymer resists or molds, including electron beam lithography (EBL)[5], deep ultraviolet lithography (DUV)[6], extreme ultraviolet lithography (EUV)[7], scanning probe lithography[8], colloidal lithography[9,10] and nanoimprint lithography[11]. The latter takes advantage of macromolecular assembly showcasing DNA modular epitaxy[12] and aligned block copolymer matrix[13]. However, of the existing nanopatterning techniques, most approaches rely on resist or mold templating for pattern transfer to the underneath material through a series of material deposition/etching and solvent processing. In addition to process complexity, a central concern is the organic residues and material compatibility with various solvents involved in the process.

To this end, the direct material writing approaches, including nanostencil lithography (NSL)[14,15] anodized aluminum oxide (AAO) templating[16,17], nanotransfer printing[18,19] and charged aerosol jets[20], have been explored. For example, NSL is a shadow-mask technique that attaches a piece of nanoaperture membrane (nanostencil) to a substrate, followed by placing the stack on top of an evaporation source in vacuum to allow material flux passing through the nanoapertures, yielding identical, or positive-tone, material deposits on the

[1]School of Integrated Circuits, Huazhong University of Science and Technology, Wuhan, China. [2]Institute for Chemical and Bioengineering, ETH Zürich, Zürich, Switzerland. [3]Department of Chemical and Materials Engineering, University of Alberta, Alberta, Canada. [4]These authors contributed equally: Shuangshuang Zeng, Tian Tian. ✉e-mail: chih-jen.shih@chem.ethz.ch

substrate. Although NSL can achieve high lateral resolution with critical dimensions (CDs) down to sub-50 nm for simple patterns such as dot arrays[21], it struggles with complex patterns, e.g., the complementary, or negative-tone, dot arrays. In addition, when writing plural material patterns, the overlay accuracy relies on micromechanical alignment of different nanostencils, which often mismatches the lateral resolution. The AAO templating basically shares similar limitations with NSL. On the other hand, the nanotransfer printing and charged aerosol jet techniques demand a restricted range of processing conditions, and as a result can only operate on a relatively limited selection of materials.

Inspired by the interference lithography underlying the interference between multiple coherent laser beams[22–25], we present the molecular-beam holographic lithography (MBHL) for direct writing complex 3D surfaces and self-aligned superlattices made by multiple materials. This approach not only inherits all advantages of NSL but also largely addresses its longstanding challenges in pattern complexity and overlay accuracy.

## Results

### Moiré interference of angular molecular beams

Figure 1 presents the concept of MBHL. To begin with, we prepared two 4-inch silicon wafers for the fabrication of nanoaperture and substrate chips. A thin (50 nm) silicon nitride ($SiN_x$) membrane was grown on the first wafer by the low-pressure chemical vapor deposition (LPCVD). Nanoapertures in the $SiN_x$ film were patterned by

electron-beam lithography, followed by selective reactive ion etching (RIE). The $SiN_x$ nanoaperture membrane was then released from the supporting substrate by a combination of dry and wet deep silicon etching through a window defined by the ultraviolet (UV) photolithography on the other side of the wafer. For the second wafer, another UV photolithography process was carried out to define the MBHL opening area by a photoresist (PR). The first and second wafers were cut to $1 \times 1\,cm^2$ and $2 \times 2\,cm^2$ square pieces, corresponding nanoaperture and substrate chips, respectively (Fig. 1a). The two chips were vertically stacked with clamps, so that the separation between the substrate chip surface and nanoaperture membrane, $D$, is precisely defined by the PR thickness (Fig. 1b). The chip stack was then placed on top of an evaporation source in a thermal or electron-beam (e-beam) evaporator to allow the evaporated material flux transferring through the nanoapertures in vacuum. More details please see "Methods".

The origin of MBHL is the generation of angular molecular beam. We noticed that the concept of glancing angle deposition (GLAD) has been used to grow nanostructured thin films[26–28] and individual nanostructures[29,30]. In our system, we refer the angular molecular beams to the GLAD material flux transferring through the nanoapertures. There are several possibilities; for example, one can incline the chip stack by a tilting angle, $\varphi$, which equals the angle between the substrate surface normal, $z$, and the material flux direction (Fig. 1b). Upon evaporation, the angular material flux passes through a nanoaperture and travels across a finite separation $D$, such that the actual landing point of a given vapor particle on the substrate surface, $(x, y)$, deviates from the incidence

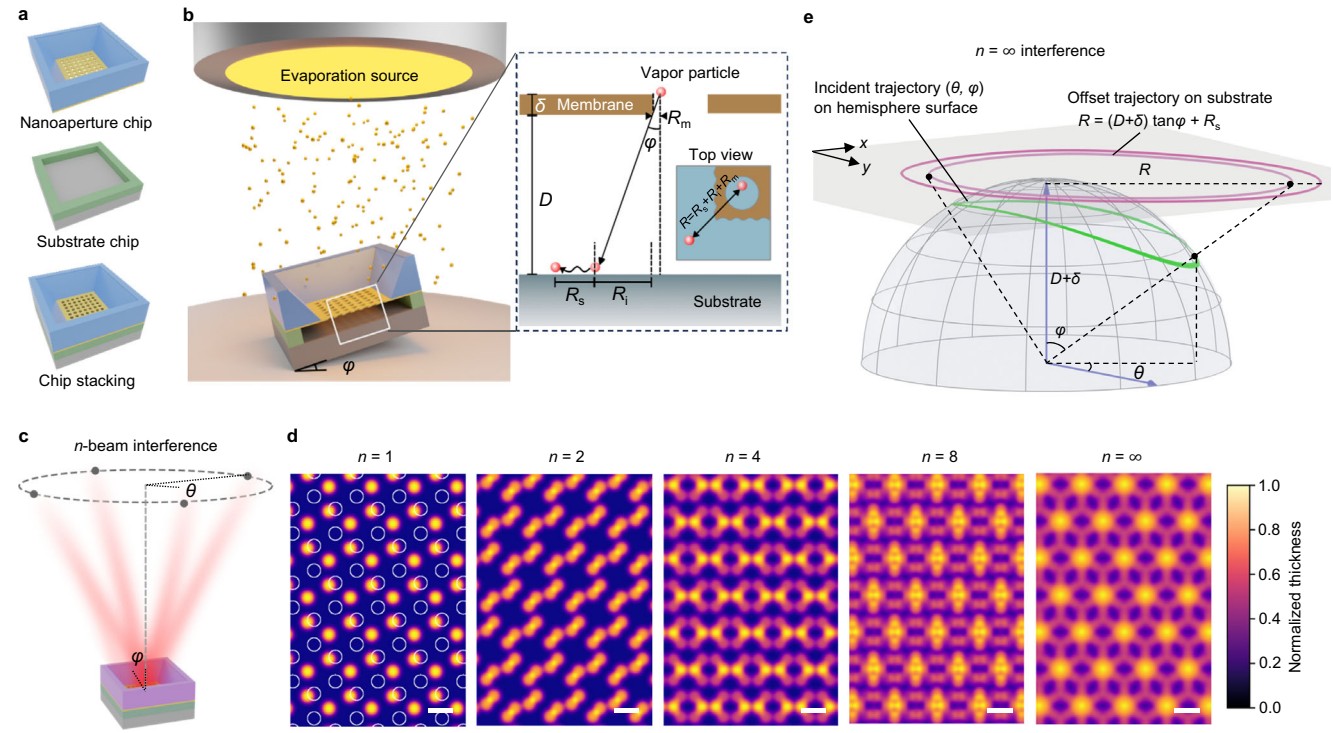

**Fig. 1 | MBHL concept. a** Schematic diagrams for the fabricated nanoaperture and substrate chips and their stack. The thickness of photoresist (green layer) on the substrate chip precisely defines the separation between the substrate surface and nanoaperture membrane, $D$. **b** Illustration for the generation of angular molecular beam when placing a tilted chip stack on top of an evaporation source. For a given vapor particle transferring through the nanoaperture, the tilting angle $\varphi$ makes the actual landing point deviate from the incidence point by a lateral offset $R = R_m + R_i + R_s = (D + \delta) \tan \varphi + R_s$. **c** Schematic illustration for an off-axis deposition system, which allows deposition from multiple coplanar evaporation sources of different $(\theta, \varphi)$ individually generating angular molecular beams superimposed on the substrate surface. Here we consider $n$ evaporation sources evenly placed on the base circumference with an identical tilting angle $\varphi$ and a common rotating

angle difference $\Delta\theta = 2\pi/n$. **d** Consider a nanoaperture pattern consisting of honeycomb lattice of circular nanopores (white circles in $n = 1$ panel). When the offset $R$ is comparable to the nanoaperture spacings, the superimposition of multiple molecular beams projected on the substrate surface yields the Moiré interference pattern. With increase of $n$, the calculated interference patterns become more and more complex, evolving from isolated hexagonal dots ($n = 1$) to continuous 3D surfaces ($n = \infty$). Scale bars, center-to-center distance $L$ of the honeycomb lattice. **e** Generalization of $n = \infty$ interference. The angular material fluxes coming from all directions $(\theta, \varphi)$ correspond to a continuous incident trajectory in the polar coordinate system (green curve), which yields the offset trajectory, a circle of variable radius $R \approx (D + \delta) \tan \varphi$, corresponding to the projection of incident trajectory to the top tangent plane (magenta curve).

point on the membrane surface $(x_m, y_m)$. The resulting lateral offset follows, $R = R_m + R_i + R_s = \sqrt{(x - x_m)^2 + (y - y_m)^2}$, where $R_m$, $R_i$, and $R_s$ correspond to the contributions of membrane, membrane-substrate gap, and surface diffusion, respectively. The geometry of the MBHL setup gives $R = (D + \delta) \tan \varphi + R_s$, where $\delta$ is the thickness of the membrane. By ignoring the surface diffusion effect, one can approximate $R \approx (D + \delta) \tan \varphi$ for large separations (see "Methods").

The principle of angular molecular beam can be further extended to the off-axis deposition systems, in which the substrate normal axis is not coincident with the center of evaporation source. The scenario adds one more degree of freedom, given the fact that the relative position vector between the chip stack and evaporation source fundamentally defines two angles, namely the rotating and tilting angles, $\theta$ and $\varphi$, respectively (Fig. 1c). Accordingly, one can carry out simultaneous or sequential deposition from multiple coplanar evaporation sources; each has different $(\theta, \varphi)$ generating individual angular molecular beams that superimpose on the substrate plane. At low tilting angles ($\varphi < 15°$), the beam shape projected on the substrate surface resembles the nanoaperture geometry due to negligible self-shadowing effect (see "Methods"), when the offset $R$ is comparable to the nanoaperture spacings and periodicities, the superimposition of multiple molecular beams projected on the substrate surface yields a Moiré interference pattern on the substrate surface.

Figure 1d presents our computational lithography (CL) simulated interference patterns considering a simple off-axis system in which $n$ evaporation sources are evenly placed on the base circumference, having an identical titling angle $\varphi$ and a common rotating angle difference $\Delta \theta = 2\pi/n$. For all subsequent discussions, the angular coordinate $\theta$ is defined to originate from the $+y$ direction and increase in the counterclockwise direction. The nanoaperture pattern design is a honeycomb lattice of circular nanopores (detailed parameters see "Methods"). For single-beam evaporation, $n = 1$, the resulting pattern basically resembles the nanoaperture pattern as expected, with an offset of distance $R$ from the original nanoaperture location. With increase of $n$, the interference patterns become more and more complex, evolving from isolated dots to continuous 3D surfaces. In particular, the $n = \infty$ interference pattern appears beyond simple Moiré fringes and cannot be fabricated by any other lithography technique in a scalable manner.

Although the $n = \infty$ case may appear theoretically complicated, it is surprisingly straightforward to implement in practice by using a rotational setup. Indeed, rather than placing infinite number of evaporation sources, we realized that angular evaporation from one single source towards a chip stack that rotates horizontally along the surface normal $z$ would lead to identical effects. We further generalized the $n = \infty$ interference technique, as illustrated in Fig. 1e. Specifically, because the separation $D$ remains constant throughout the process, the movement of continuously revolving chip stack, described by two parameters $(\theta, \varphi)$, is equivalent to having an infinite number of evaporation sources placed along a continuous trajectory on the hemisphere with radius $D + \delta$. From the perspective of the chip stack, the collective incidence of angular material fluxes coming from all directions $(\theta, \varphi)$ reaching a point at the nanoaperture can be represented as a continuous trajectory on the surface of hemisphere of radius $D + \delta$. The actual offset trajectory writing on the substrate surface, which is a circle of variable radius $R \approx (D + \delta) \tan \varphi$, corresponds to the orthogonal projection to a tangent plane $(x, y)$ placed at the celestial pole. Our CL algorithm is built on this principle, as will be discussed in detail later. In the following sections, we will demonstrate the capability of MBHL with increasing level of complexity.

## One-dimensional interference patterns

We first discuss the one-dimensional (1D) interference patterns, where the Moiré interference results from 1D displacement of molecular

beams. The most straightforward implementation is the $n = 2$ interference in an off-axis system (Figs. 1c, 2a), with the left and right beams generating a symmetric offset, $\pm R$. Figure 2a presents an aerial-view illustration for a 3D surface formed by the superimposition of two angular molecular beams passing through a ring-shaped nanoaperture. The deposits in the intersect of two rings appear to accumulate more amount of material, forming a wavy 3D surface. By making use of this effect, Fig. 2b presents a number of Moiré interference patterns made by gold (Au), with precisely controlled values of $R$, by evaporating through a bullseye nanoaperture. To shed light on the Moiré interference effect, we compare the $n = 1$ and $n = 2$ 3D surfaces made by a guest:host organic semiconductor system, Ir(ppy)$_3$:CBP (4,4′-bis($N$-carbazolyl)-1,1′-biphenyl), as revealed in their phosphorescence and atomic force microscopy (AFM) height images (Fig. 2c, d). The MBHL technique presented here represents the only approach allowing direct 3D nanopatterning of the solvent-sensitive organic semiconductors.

Another interesting set of 1D interference patterns is the $n = \infty$ interference for the evaporation through multiple parallel slits of width $s_1$ and spacing $s_2$ (Fig. 2e and Supplementary Fig. 1). When $s_2 \gg R$, the evaporation through an individual slit results in a "cat-head" 3D line surface (Fig. 2e bottom). With decreasing $s_2$, the resulting patterns through individual slits interact more strongly with each other, leading to more complex 3D surfaces beyond the classical Moiré interference. In combination with our CL simulations (see "Methods" and Supplementary Fig. 2), we constructed a "phase diagram" representing different groups of 3D surfaces with respect to two dimensionless parameters, $\lambda_1 = s_1/(s_1 + s_2)$ and $\lambda_2 = s_1/(s_1 + 2R)$, informing the slit-to-spacing and slit-to-offset ratios, respectively, for the generalization of our experimental results (Fig. 2f). We identify four major groups of 3D surfaces resulting from the slit interference, including: (I) the fully separated patterns, $\lambda_2 > \lambda_1$, (II) the sawtooth patterns, $\lambda_1 > \lambda_2 > \lambda_1/(\lambda_1 + 1)$, (III) the comb patterns, $\lambda_1/(\lambda_1 + 1) > \lambda_2 > \lambda_1/2$, and (IV) the high-order interference patterns, $\lambda_2 < \lambda_1/2$. We fabricated a series of 3D line surfaces made by germanium (Ge) on the $(\lambda_1, \lambda_2)$ landscape. Our CL simulated topography nicely describes the AFM height profiles (Fig. 2f insets).

## Two-dimensional interference patterns

With a proper design of nanoaperture pattern and offset trajectory, the material deposited at a given position $x = (x, y)$ on the substrate plane can come from angular beams of varied azimuthal and polar angles $(\theta, \varphi)$, which yields the two-dimensional (2D) interference patterns. To elucidate the basic principle, we will explain our CL algorithm in depth, using the $n = \infty$ interference as an example. Specifically, following our discussion in Fig. 1e, the incidence of angular beams coming from different angles leads to an offset trajectory writing on the substrate plane, $\mathbf{F}(\mathbf{x})$. Similarly, the nanoaperture pattern is described by a 2D binary function $\mathbf{M}(\mathbf{x})$, following $\mathbf{M}(\mathbf{x}) = 1$ for $\mathbf{x} \in \Omega$ and $\mathbf{M}(\mathbf{x}) = 0$ for $\mathbf{x} \notin \Omega$, where $\Omega$ is the nanoaperture domain. In practical MBHL setups where the self-shadowing effect of the membrane wall can be ignored (see "Methods"), the deposition probability at the given position on the substrate plane, $\mathbf{P}(\mathbf{x})$, is given by:

$$P(x) = M(x) \otimes F(x) \qquad (1)$$

where the symbol $\otimes$ denotes the 2D convolution operation (see "Methods"). As the material flux and deposition conditions remain uniform, the height profile $\mathbf{H}(\mathbf{x})$ can be effectively approximated by $\mathbf{P}(\mathbf{x})$ under these conditions (see "Methods"). Indeed, as revealed in Fig. 3a, when the offset trajectory function intersects four nanopores in the nanoaperture pattern, $\mathbf{M} \otimes \mathbf{F} > 0$, the material deposited at a given position $\mathbf{x}$ come from all four nanopores, leading to the "constructive interference". On the other hand, if the offset trajectory does

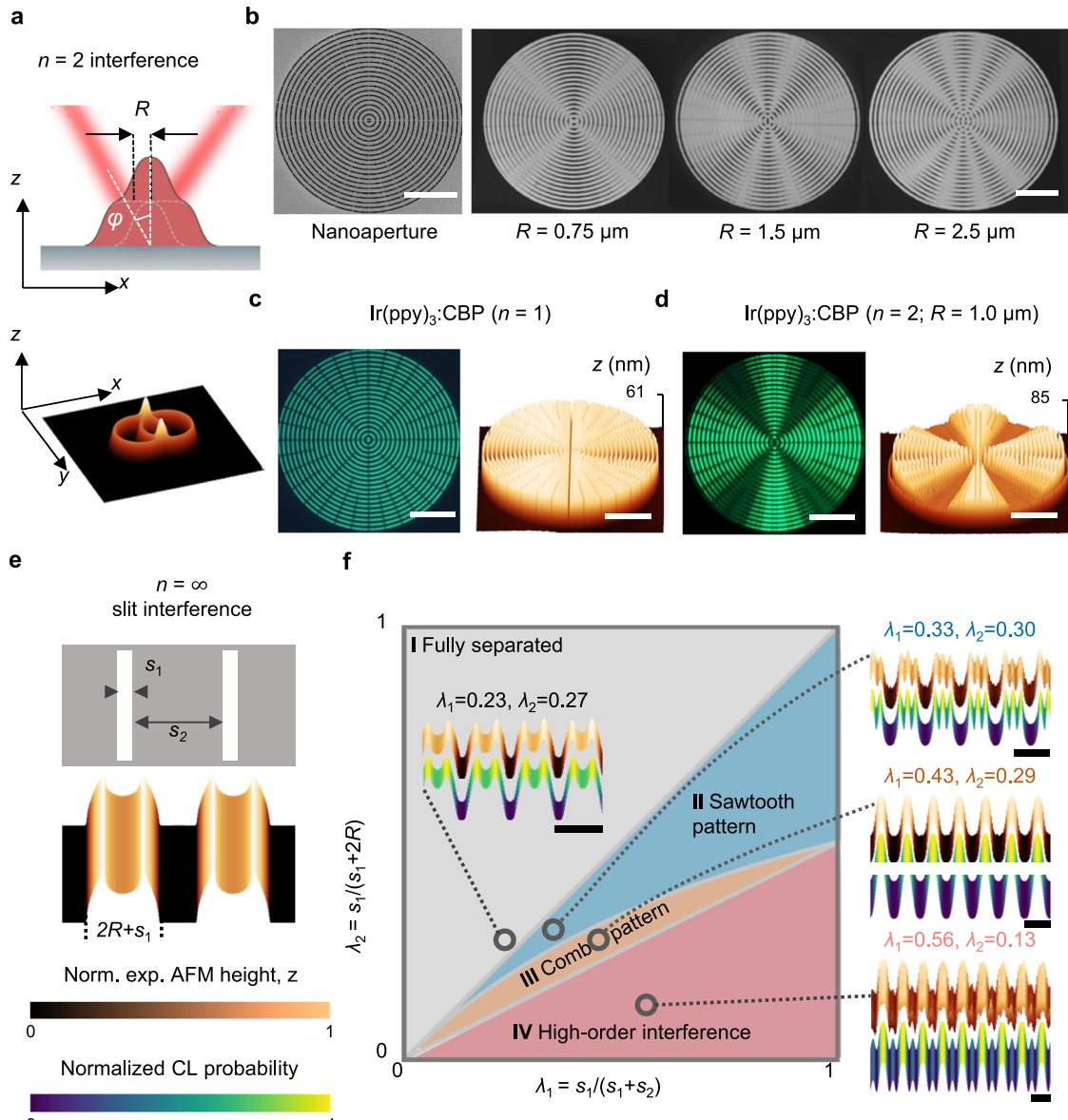

**Fig. 2 | Demonstration of 1D interference patterns. a** Schematic illustration for an $n = 2$ interference pattern in an off-axis system, with a symmetric beam offset, $\pm R$. The superimposition of two angular beams passing through a ring-shaped nanoaperture forms a 3D surface with a larger amount of material deposited at the intersect of two rings. **b** Scanning electron microscope (SEM) images for a bullseye nanoaperture and the fabricated Moiré interference patterns made by Au, with precisely controlled values of $R$. Scale bars, 10 μm. **c, d** Comparison for phosphorescence (left) and AFM height (right) images of $n = 1$ (**c**) and $n = 2$ (**d**) 3D surfaces made by Ir(ppy)$_3$:CBP. Scale bars, 10 μm. **e, f** Understanding $n = \infty$ slit interference patterns formed by evaporating Ge through multiple parallel slits of width $s_1$ and spacing $s_2$, which forms a "cat-head" 3D line surface for $s_2 \gg R$ (**e**). With decrease of $s_2$, the interaction between material wavefronts lead to more complex 3D surfaces, which can be mapped on a "phase diagram" constructed with respect to two dimensionless parameters, $\lambda_1 = s_1/(s_1 + s_2)$ and $\lambda_2 = s_1/(s_1 + 2R)$. Four major groups of 3D surfaces are identified, as revealed by the experimentally measured (Exp.) AFM topography for the fabricated 3D line surfaces and corresponding CL simulations. Scale bars, $(s_1 + s_2)$ (**f**). The colors in each figure indicate the intensity of normalized height $z$ from experimental AFM measurements or normalized probability from CL model.

not fall into the nanoaperture domain, $\mathbf{M} \otimes \mathbf{F} = 0$, no material is deposited at $\mathbf{x}$, namely the "vacant interference".

Figure 3b further illustrates the geometric relationships on constructive and vacant interference for nanoaperture patterns of square and hexagonal nanopore lattices, with pore radius $r$ and center-to-center distance $L$. We consider the process conditions similar to the results of $\varphi = 1°$ in Supplementary Fig. 3, where $D$ and $\varphi$ remain constant and $\theta$ continuously varies from 0 to $2\pi$. As the offset trajectory pattern $\mathbf{F}$ is a circle of constant radius $R \approx (D + \delta) \tan \varphi$ (see Fig. 1e), the deposition probability at the position directly underneath the center of a nanopore, $P_c$, is proportional to the total length of individual intersecting arcs, $C_i$, between the circle $\mathbf{F}$ and the $i$-th nearest

nanopores. Accordingly, we have generalized the expression of $P_c$ as follows (see "Methods"):

$$P_C = \sum_{i=1}^{\infty} \frac{N_i}{\pi} \text{Re} \left\{ \cos^{-1} \left[ \frac{\left(\frac{R}{r}\right)^2 + \left(\frac{L_i}{r}\right)^2 - 1}{2 \frac{R}{r} \frac{L_i}{r}} \right] \right\} + \varkappa (r - R) \qquad (2)$$

where $i$ is the index of neighboring nanopores, Re is the real part of a complex number, $\varkappa$ is the Heaviside step function, $N_i$ and $L_i$ are the number of equivalent nanopores and center-to-center distance of $i$-th neighbors, respectively. For example, for the circular trajectories considered in Fig. 3b, in which the constructive interference takes

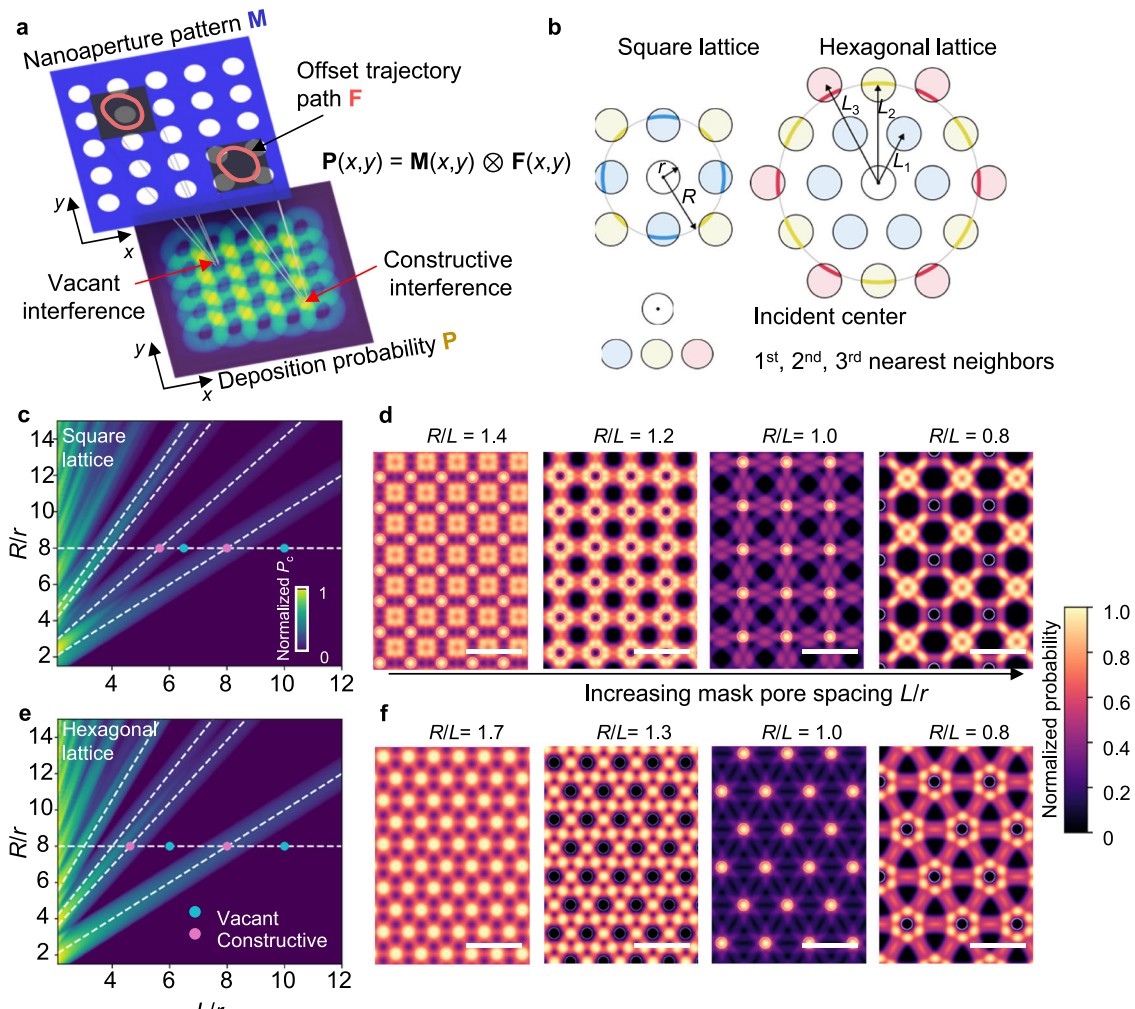

**Fig. 3 | Illustration of computational lithography algorithm by elucidating the 2D constructive and vacant interference in $n = \infty$ interference patterns.**
**a** Schematic diagram illustrating the probability for material deposited at a $\infty$ given position $\mathbf{x}(x,y)$ on the substrate plane is proportional to the 2D convolution of the offset trajectory function $\mathbf{F}(x,y)$ and the nanoaperture pattern function $\mathbf{M}(x,y)$.
**b** Geometric relationship for the constructive interference at the position directly underneath the center of a nanopore, in which the angular beams come from the 1st and 2nd nearest nanopores for the square lattice and the 2nd and 3rd nearest nanopores for the hexagonal lattice. **c**–**f** Dimensional analysis for the deposition probability at the nanopore center position, $P_c$, as a function of two dimensionless variables, $R/r$ and $L/r$, for square (**c**) and hexagonal (**e**) nanopore lattices, revealing that the constructive interference takes place along the lines of $R/L = c_i$, where $c_i = \{1, \sqrt{2}, 2, \sqrt{5}, \ldots\}$ and $c_i = \{1, \sqrt{3}, 2, \sqrt{7}, \ldots\}$ for square and hexagonal nanopore lattices, respectively. Along a given horizontal cut, $R/r = 8$, we present a number of simulated 2D interference patterns corresponding to different $R/L$ values for square (**d**) and hexagonal nanopore lattices (**f**). The white circles denote the nanopores location. The colors in each figure indicate the intensity of normalized probability $P_c$ from CL model. Scale bars: 10$r$.

place at the nanopore center position, the angular beams come from the 1st and 2nd nearest nanopores for the square lattice and the 2nd and 3rd nearest nanopores for the hexagonal lattice. On the other hand, depending on the relative ratio between $R$ and $L$, Eq. (2) suggests that the vacant interference emerges when the offset trajectory no longer intersects with any nanopores.

Figure 3c, e present the dimensional analysis of Eq. (2) as a function of two dimensionless variables, $R/r$ and $L/r$, for square and hexagonal nanopore lattices, respectively, revealing the landscape of constructive (high probability) and vacant (low probability) interference. Interestingly, the maximum ridges of constructive interference coincide with the straight lines for a series of constant parameters $R/L = c_i$, where $c_i = \{1, \sqrt{2}, 2, \sqrt{5}, \ldots\}$ and $c_i = \{1, \sqrt{3}, 2, \sqrt{7}, \ldots\}$ for square and hexagonal nanopore lattices, respectively. More analysis please visit Methods and Supplementary Figs. 3, 4. Along the horizontal cuts corresponding to constant $R/r$ values, we have observed alternating constructive and vacant interference regimes, which echo the simulated 2D interference patterns (Fig. 3d, f) of various values of parameter $R/L$.

We noticed that the deposition material plays an important role in determining the interference pattern owing to different degrees of lateral diffusion. For example, Fig. 4a compares the fabricated interference patterns made by silicon oxide (SiO$_2$) and Ge using the same nanoaperture design of square and hexagonal lattices ($r = 50$ nm and $L = 500$ nm). The average CL-fitted values of surface diffusion length, $R_s$, are 70 and 152 nm, respectively. Indeed, the tangential component for an angular molecular beam vector results in a directional drift on the surface, such that such that $R_s = R_d + R_f = N(R_d, \sigma_f)$, where $R_d$ and $R_f$ are the directional and non-directional contributions to the surface diffusion length, respectively, $\mathcal{N}$ is the normal distribution function, and $\sigma_f$ is the variance for the non-directional surface diffusion (see "Methods"). The directional surface diffusion originates from the conservation of lateral momentum for the incident particles approaching the surface[31]. If the particle-surface interaction is weak, the incident particle can travel farther before being immobilized by surface trapping sites or defects[19], resulting in a long $R_d$. The precision of MBHL relies on the reproducibility of $R_d$ on different surface

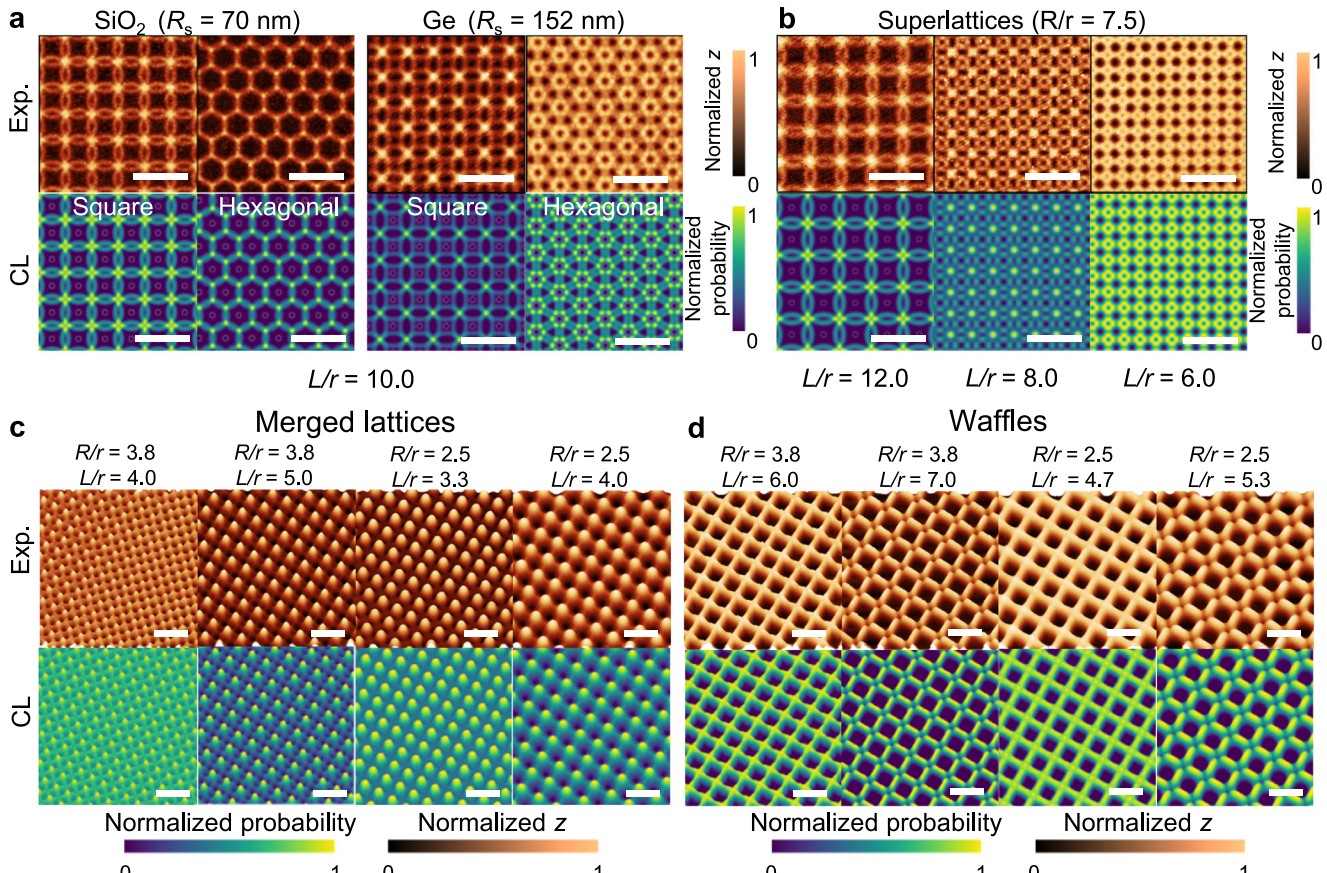

**Fig. 4 | Demonstration of 2D interference patterns. a** Comparison of fabricated 3D surfaces made by SiO$_2$ and Ge using identical nanoaperture design of square and hexagonal nanopore lattices ($r = 50$ nm and $L/r = 10.0$). The average fitted values of surface diffusion length, $R_s$, are 70 and 152 nm, respectively. **b–d** 2D interference patterns made by Ge using nanoapertures of square nanopore lattice in the condition of $D = 2.5$ μm and $\varphi = 5°$. $R \approx (D + \delta)\tan\varphi + R_s = 375$ nm. Upon variation of $r$

and $L$, the experimentally measured (Exp.) AFM height (top row) and CL simulated probability (bottom row) profiles reveal three groups of 3D surfaces: (I) the complex lattice patterns (**b**), (II) the merged lattice patterns (**c**), and (III) the waffle patterns (**d**). The colors in each figure indicate the intensity of normalized height $z$ from experimental AFM measurements or normalized probability from CL model. Scale bars: 1 μm.

conditions. We examined the radii of deposited ring patterns made by different materials, which suggest the ranking of $R_s$ for various materials is as follows: Au ≈ Ge > SiO$_2$ > Al$_2$O$_3$ ≈ Ag (Supplementary Fig. 5).

The incorporation of surface diffusion effect into our CL model allows us to nicely describe the 3D morphology of the fabricated interference patterns. Figure 4b–d presents a number of 2D interference patterns comparing between the experimentally measured AFM height and CL simulated probability profiles. All the fabricated 3D surfaces were based on the Ge deposition through simple nanoapertures of square nanopore lattice. Analogous to the dimensional analysis presented in Fig. 2f, we expect there exists a multi-dimensional "phase diagram", which represents different groups of 3D surfaces with respect to multiple dimensionless parameters, for the generalization of all possible morphologies based on the square nanopore lattice. By varying $r$ and $L$ in the nanoaperture design (Supplementary Fig. 6), we have successfully identified three groups of 3D surfaces, including: (I) the complex lattice patterns (Fig. 4b), (II) the merged lattice patterns (Fig. 4c), and (III) the waffle patterns (Fig. 4d). In particular, the group (III) patterns arise from vacant interference beneath the nanopores, forming the purely negative-tone patterns relative to the nanoaperture design. The generated patterns in group (II) and (III) displayed unique optical features in the reflection spectrum (Supplementary Fig. 7), as simulated using commercial Ansys Lumerical software.

### Self-aligned superlattices

In lithography, overlay control, which is about precise alignment between patterns fabricated by different mask layers, is known as one of the most critical yield limiters[32]. A key advantage for the MBHL technique presented here is to allow superimposition of multiple 3D surfaces made by different materials without changing the relative position between the substrate and nanoaperture chips. Indeed, consider the overlay of various offset trajectories writing on the substrate, since a typical accuracy for mechanical goniometer is $\Delta\varphi < 0.1°$, one can readily estimate the overlay accuracy between different layers of interference patterns, $\Delta R$, at low angles is approximately given by $\Delta R \approx (D + \delta)\tan(\Delta\varphi) \approx (D + \delta)\Delta\varphi$. Given a typical separation $D$ of ≈1 μm, it follows $\Delta R < 2$ nm, which is comparable to the state-of-the-art TWINSCAN® DUV/EUV systems[33], without involving sophisticated laser interferometry. The above rationale motivated us to explore the self-aligned superlattices superimposing multiple interference patterns (Fig. 5). All superlattice patterns were fabricated based on the square nanopore lattices.

Based on our theoretical analysis in Fig. 3c about constructive and vacant interference underneath the center of a given nanopore in square lattice, we first examined the superlattices superimposing one vacant and one constructive interference patterns, as shown Fig. 5a. Each superlattice is prepared in single run without breaking the vacuum by changing the e-beam evaporation sources in sequence. The vacant interference patterns are made by Ag following the green incident trajectory on the $(\theta, \varphi)$ polar coordinate, resulting in the

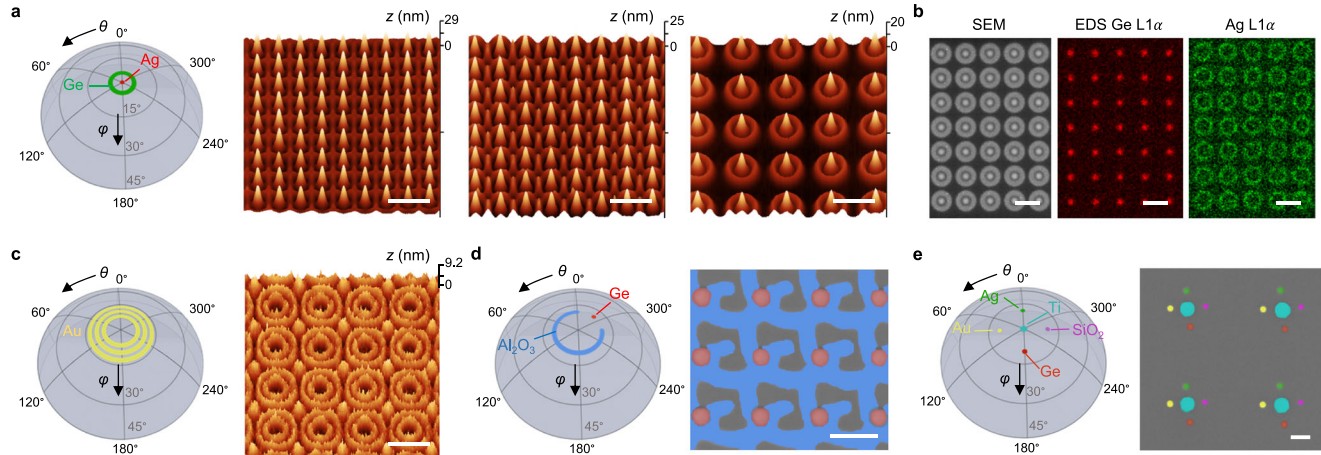

**Fig. 5 | Demonstration of self-aligned superlattice patterns based on the square nanopore lattices. a** Superlattices that superimpose a vacant interference pattern by depositing Ag along the incident trajectory (green circle) of $\varphi = 5°$ and a constructive pattern by perpendicular deposition of Ge (red dot). The AFM height images present superlattices deposited through nanoapertures of square nanopore lattices, with $r = 75$ nm and $L = 550$, 650, and 1150 nm (left to right). **b** SEM and EDS mapping images that clearly differentiate Ag and Ge distributions in the superlattice of $r = 75$ nm and $L = 1150$ nm. **c** A flower-shape 3D surface made by Au. The superimposition of three vacant interference patterns were fabricated using three isolatitude incident trajectories at $\varphi = 7°$, $10°$, and $13°$. **d** False-color SEM image showing a symmetry-breaking superlattice pattern fabricated by the depositions of Al$_2$O$_3$ along the blue incident trajectory ($\varphi = 10°$; $\theta = 0$ to $270°$) and Ge at the red incident angle $(\theta, \varphi) = (315°, 10°)$. **e** False-color SEM image showing a quinary chiral superlattice comprised of 5 "meta-atoms" made by Ag, Au, Ge, SiO$_2$, and Ti. The first four materials were deposited at incident angles $(\theta, \varphi)$ of $(0°, 10°)$, $(90°, 10°)$, $(180°, 10°)$, and $(270°, 10°)$, respectively. The Ti pattern was deposited along an isolatitude incident trajectory at $\varphi = 0.5°$. Scale bars in all images, 1 μm.

waffle (group (III) in Fig. 4d) or ring-shaped patterns. Subsequently, the cone-shaped patterns made by Ge, in which individual cones perfectly align with the void centers, were fabricated using a perpendicular deposition, corresponding to the incidence at the celestial pole. The material composition distribution is imaged by the energy dispersive spectrum (EDS) mapping (Fig. 5b). The overlay accuracy between the ring and cone interference patterns is determined to be around 2 nm (Supplementary Fig. 8). More systematic results by varying the nanoaperture pattern design please see Supplementary Fig. 9.

Figure 5c presents a flower-shape 3D surface by superimposing three vacant interference patterns fabricated by using three isolatitude incident trajectories. The last trajectory results in a merged lattice pattern (group (II) in Fig. 4c). Apart from the 2D symmetric superlattices, we also demonstrated a number of chiral symmetry-breaking patterns made by multiple materials, as shown in Fig. 5d, e, together with their incident trajectory designs. In particular, Fig. 5e presents a quinary superlattice made by 5 materials, Ag, Au, Ge, SiO$_2$, and Ti, in which individual "meta-atoms" are arranged in a self-aligned manner. The smallest dot and line structures fabricated using MBHL show ≈ 57 nm in diameter and 47 nm in width, respectively, as shown in Supplementary Fig. 10.

## Discussion

Beyond the ultra-fine resolution, the power of MBHL lies in its vast design space for beam trajectories. Theoretically, MBHL is capable of approximating any 2D periodic design $\mathbf{P}(\mathbf{x})$ by solving the optimal trajectory function $\mathbf{F}(\mathbf{x})$ from the deconvolution of Eq. (1). Although the exact inverse design poses challenges due to practical constraints from deposition systems, here we demonstrate a few examples of advanced patterns formed using few-beam MBHL interference. In the simplest $n = 2$ interference illustrated in Fig. 2, where the azimuthal angles ($\theta$) for the beam trajectory are $\{\theta_0, \theta_0 + \frac{\pi}{2}\}$, the angle displacement $\theta_0$ strongly influences the interaction between neighboring deposits. Figure 6a–e show the spectrum of pattern morphologies achievable by adjusting $\theta_0$ on a square lattice nanoaperture with pore radius $r$, center-to-center spacing $L$ and beam offset $R$. Inspired by the ranking of periodic nanomaterials' dimensionality[34] (see "Methods"), we created a comprehensive phase diagram of the pattern

morphology as a function of the parameters $r/L$, $R/L$, and $\theta_0$, ranging from 0D (isolated particles) to 1D (stripes) and 2D (planar) geometries. Supplementary Fig. 11 demonstrates a similar phase diagram for hexagonal nanoapertures under such conditions. We also anticipate that a misalignment between the axis of mirror symmetry of the trajectory function and that of the nanostencil will lead to the formation of a geometrically chiral pattern. Figure 6f, g demonstrate the evolution of the geometric chirality in the MBHL deposition patterns for $n = 3$ interference on honeycomb nanoapertures ($\theta \in \{\theta_0, \theta_0 + \frac{\pi}{3}, \theta_0 + \frac{2\pi}{3}\}$, top row) and $n = 4$ interference on square nanoapertures ($\theta \in \{\theta_0, \theta_0 + \frac{\pi}{4}, \theta_0 + \frac{\pi}{2}, \theta_0 + \frac{3\pi}{4}\}$, bottom row). By leveraging the concept of geometric overlap measure for planar chiral surfaces[35], we employ a chirality index $S$ to evaluate the chirality of the formed surface patterns (see "Methods"):

$$S = \min_{u,v,\tau} \frac{1}{|D|} \iint_D |P_+(x) - T(u,v,\tau)P_-(x)| dx \quad (3)$$

where $P_+$ and $P_-$ are the mirror images of the deposition patterns on domain $D$, and $T(u,v,\tau)$ denotes the translation by vector $(u,v)$ followed by rotation of $\tau$. For an achiral surface, $S = 0$, whereas $S = 1$ indicates a perfectly chiral surface. As shown in Supplementary Figs. 12 and 13, the continuous modulation of the geometric chirality of nanosurfaces can be achieved by simply tuning the angle displacement of the beam trajectory, which is otherwise a non-trivial task in other fabrication processes[36]. Our numerical optimization via the CL model indicates that the maximum $S$ values for the honeycomb (Fig. 6f) and square (Fig. 6g) nanoapertures are 0.46 and 0.57, respectively, manifesting as propeller and gammadion, respectively. The capacity of the MBHL process for precise manipulation of overlap, dimensionality, and chirality offers vast potential for applications in nanophotonic and plasmonic devices. We foresee further innovative trajectory designs emerging from future research, expanding the functional scope of MBHL.

In conclusion, we have demonstrated a direct nanopatterning method for the fabrication of complex 3D surfaces and self-aligned superlattices based on the Moiré interference of molecular beams. The MBHL is in-principle capable of making any periodic complex patterns

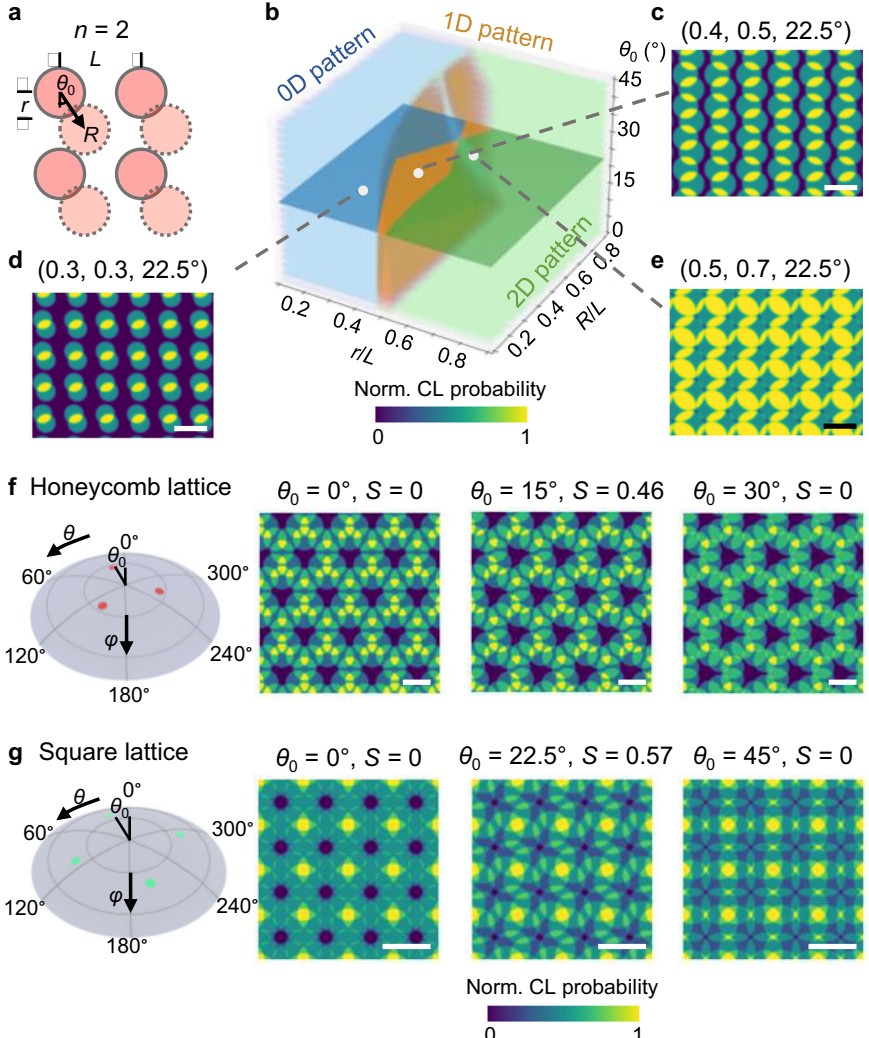

**Fig. 6 | Advanced pattern design based on MBHL. a** Diversity of deposition morphology formed by angle-misaligned $n = 2$ interference. **b** Phase diagram as a function of dimensionless parameters $\frac{r}{L}$, $\frac{R}{L}$, and $\theta_0$ which distinguishes between 0D (blue), 1D (orange), and 2D (green) patterns. **c–e** Three representative CL-simulated patterns for 0D (isolated particles), 1D (stripes) and 2D (planar structures) regimes on the horizontal cut where $\theta_0 = 22.5°$. Scale bars, $L$ (center-to-center spacing). **f–g** Evolution of pattern chirality by tuning the beam angle displacement $\theta_0$. Two examples of $n = 3$ interference on honeycomb lattice nanoaperture (**f**) and $n = 4$

interference on square lattice nanoapertures (**g**) are demonstrated. When the axis of mirror symmetry for the trajectory pattern aligns with that of the nanoaperture ($\theta_0 = 0°$ and $\theta_0 = 30°$ for honeycomb lattice and $\theta_0 = 0°$ and $\theta_0 = 45°$ for square lattice, respectively), the formed patterns are achiral ($S = 0$). The patterns with maximal geometric chirality are obtained at $(r/L, R/L, \theta_0) = (0.45, 0.68, 15°)$ for honeycomb lattice, with an $S$-value of 0.46 and $(r/L, R/L, \theta_0) = (0.41, 0.54, 22.5°)$ for square lattice, with an $S$-value of 0.57, respectively. Scale bars, $L$.

of translational symmetry. A high degree of symmetry-breaking and/or chirality can be introduced by the conjugation of incident trajectory, nanoaperture design, and materials combination. This technology is particularly useful for patterning solvent-sensitive materials, such as ionic electronic materials and organic semiconductors. For example, an exciting technological opportunity is to use MBHL for the fabrication of dielectric or plasmonic nanoresonators/nanocavities surrounding nanoscale pixels of organic semiconductor devices, such as light-emitting diodes and photodiodes. In general, thanks to the nanoscale critical dimensions and overlay accuracy offered by MBHL, we anticipate the method will facilitate large-scale fabrication of devices for a broad range of applications, such as nanoimaging, molecular sensing, catalysis, and optoelectronics.

## Methods

### SiN$_x$ nanoaperture fabrication
The SiN$_x$ nanoapertures were fabricated on 4-inch silicon wafers with a 50nm-thick low stress SiN$_x$ layer on both sides deposited by LPCVD.

Firstly, nanoapertures were patterned in the SiN$_x$ layer on the front side of the wafer by electron beam lithography (EBPG5200, Raith) and followed with RIE (PlasmaPro 80 RIE, Oxford Instruments). Windows were then opened on the backside of the wafer by conventional UV lithography (EVG 620 NT, EV Group) with alignment to the nanoapertures on the front side. The bulk silicon was etched through the opened windows by deep reactive ion etching (SPTS) and KOH wet etching to release the membranes on the front side. After membrane release, the wafers were cleaved into 1 cm × 1 cm chips for the nanopatterning process.

### Substrate preparation
Substrates with spacer were prepared using UV lithography. A bare silicon wafer was first cleaned with acetone and isopropanol in sequence. Then PR was spun coated at 4000 rpm and baked on a hot plate at 110 °C for 2 min. Different spacer heights were defined using different PR. In specific, AZ1505 and AZ1518 (MicroChemicals) were used to define 0.7 μm and 2.5 μm spacer height separately. After

exposure and development, the wafer was cleaved into 2 cm × 2 cm chips for the nanopatterning process.

## Deposition conditions and characterization

Ag (99.99%), Au (99.99%), Ge (99.999%), Ti (99.995%), SiO$_2$ (99.99%) and Al$_2$O$_3$ (99.99%) were evaporated in a commercial electron beam evaporation system (MEB 550S, Plassys) at <5×10$^{-7}$ mbar. The deposition rate was 0.5 nm s$^{-1}$ for the various materials. Guest:host organic semiconductor system, Ir(ppy)$_3$ (>99%):CBP (4,4'-bis(N-carbazolyl)-1,1'-biphenyl) (>99%), was evaporated in a custom-built high-vacuum thermal evaporation chamber inside the glove box. The deposition rate was 0.1 nm s$^{-1}$. Scanning electron microscopy (SEM) images were acquired using a field emission microscope (Ultra plus, Zeiss). Electron dispersive X-ray (EDX) mapping results were obtained using a Zeiss Ultra 55 equipped with a Ultim Max detector (Oxford Instruments). Atomic force microscopy (AFM) images were carried out on NX20 (Park Systems) in non-contact mode. Phosphorescence images were taken with an inverted microscope (Eclipse Ti2-U) equipped with a 50× dry objective (TU Plan Fluor, NA = 0.8, WD = 1.00 mm).

## Optical simulation

For optical simulations, the three-dimensional finite-difference time-domain (FDTD) method (FDTD Solutions, Ansys Lumerical 2023 R2) was employed to simulate the optical response of the nanostructure. The substrate was set to silicon (Si), and the nanostructure material was germanium (Ge). For all simulations, a plane wave source was used to excite the nanostructures. The FDTD calculation region was defined based on the periodicity of the samples, with Bloch boundary conditions applied along the X and Y axes. The region extended 3 μm in the Z direction, with perfectly matched layers (PML, 12 layers) at the boundaries. An analysis group of gratings was positioned 1.25 μm above the surface to record the optical response.

## Phase diagram boundaries for one-dimensional interference patterns

Supplementary Fig. 2a,b demonstrate the relative positions between two neighboring "cat-head" line wavefronts corresponding to the four major regimes in the one-dimensional interference phase diagram as shown in Fig. 2f. The boundary cases between regimes I-II, II-III and III-IV are satisfied by:

I-II Boundary: The two neighboring "cat-head" line wavefronts are touching, i.e., $s_1 + s_2 = 2(\frac{1}{2}s_1 + R) = s_1 + 2R$:

$$\lambda_1 = \frac{s_1}{s_1 + s_2} = \frac{s_1}{s_1 + 2R} = \lambda_2 \qquad (4)$$

II-III Boundary: The outermost regions in the neighboring line wavefronts overlap, i.e. $2s_1 + s_2 - R = s_1 + R$:

$$\frac{1}{\lambda_2} = \frac{s_1 + 2R}{s_1} = \frac{2s_1 + s_2}{s_1} = \frac{1}{\lambda_1} + 1 \qquad (5)$$

III-IV Boundary: The neighboring line wavefronts are overlapped in half, i.e. $s_1 + s_2 = \frac{1}{2}s_1 + R$:

$$\lambda_1 = \frac{s_1}{s_1 + s_2} = \frac{s_1}{\frac{1}{2}(s_1 + 2R)} = 2\lambda_2 \qquad (6)$$

While the current discussion is limited to four regimes, it is expected there are finer separations within the higher-order interference regime (IV).

## Derivation of Eq. (1)

Based on the general geometry of the MBHL system, as shown in Supplementary Figs. 14 and 15, for an incoming particle along the continuous trajectory **F** to reach point **x** = (x, y) on the substrate surface, it must follow the procedure shown below:

1. The location of incidence at the top of the nanoaperture membrane plane, $\mathbf{x}^T = (x_m^T, y_m^T)$ must be within regime **Ω** (i.e., inside a hole where $\mathbf{M}(x_m^T, y_m^T) = 1.$)
2. When the particle leaves the lower plane of the membrane, its exit location $\mathbf{x}^B = (x_m^B, y_m^B)$ must also be within regime **Ω** (i.e., $\mathbf{M}(x_m^B, y_m^B) = 1$)
3. The exist location $\mathbf{x}^B$ must be from the same hole region as $\mathbf{x}^T$
4. After the particle arrives at the substrate surface $\mathbf{x}^S = (x^S, y^S)$, the surface diffusion causes it to further drift a distance $R_s$ along the vector $\mathbf{u} = (x, y) - (x_m, y_m)$

Note that conditions 2 and 3 count for the "self-shadowing" effect of membrane with non-negligible thickness, where the particle cannot penetrate the stencil beyond a certain $\varphi$ angle. Using the geometry shown in Supplementary Fig. 14b, the location of $\mathbf{x}^B$ is essentially $\mathbf{x}^B = \mathbf{x} - (R_s + R_i)\hat{\mathbf{u}} = \mathbf{x} - (1 - \lambda)\mathbf{u}$, where $\hat{\mathbf{u}}$ is the unit vector of $\mathbf{u}$, and $\lambda = \frac{R_m}{R} = \frac{\delta \tan\varphi}{(D+\delta)\tan\varphi + R_s}$ is the correction factor considering the finite thickness of the membrane wall. To fulfill condition 3, we first segment the stencil function **M** into a 2D label function $\mathcal{L}$ to count individual holes, such that

$$\mathcal{L}(\mathbf{x}) = \begin{cases} 1, 2, \ldots k \ldots, K, & \text{if } \mathbf{x} \text{ is within } k - \text{th hole} \\ 0, & \text{if } \mathbf{x} \text{ is outside } \Omega \end{cases} \qquad (7)$$

To ensure that $\mathbf{x}^T$ and $\mathbf{x}^B$ are within the same hole region, their corresponding labels $l^T$ $l^B$ must be identical. Combining all above conditions, for a uniform material flux $\Phi_0$ from the source, the material flux arriving at point **x**, **Φ**(**x**), can be expressed as:

$$\mathbf{\Phi}(\mathbf{x}) = \iint_{-\infty}^{\infty} \Phi_0 w(\mathbf{u})\mathbf{F}(\mathbf{u})\mathbf{M}(\mathbf{x} - \mathbf{u})\mathbf{M}(\mathbf{x} - (1-\lambda)\mathbf{u})\Delta_{l^T, l^B} d\mathbf{u} \qquad (8)$$

where $w(\mathbf{u})$ represents the weight for each deposition event associated with vector **u**, and $\Delta_{l^T, l^B}$ is the Kronecker delta of labels $l^T$ and $l^B$. While Eq. (8) describes all the factors leading to the deposition pattern in MBHL, in most $n = \infty$ deposition cases, it can be simplified due to i) the membrane thickness is negligible compared to the membrane-substrate gap, $\frac{\delta}{D} \ll 1$, and ii) the stage rotation speed is uniform, $w(\mathbf{u}) = 1$. Most notably, the self-shadowing effect can be eliminated when $\lambda$ approaches 0. Since **M** is a binary function $\lim_{\lambda \to 0} \mathbf{M}(\mathbf{x} - \mathbf{u})\mathbf{M}(\mathbf{x} - (1-\lambda)\mathbf{u}) = \mathbf{M}^2(\mathbf{x} - \mathbf{u}) = \mathbf{M}(\mathbf{x} - \mathbf{u})$. Under these assumptions, the deposition probability $\mathbf{P}(\mathbf{x}) = \mathbf{\Phi}(\mathbf{x})/\Phi_0$ follows:

$$\mathbf{P}(\mathbf{x}) = \iint_{-\infty}^{\infty} \mathbf{F}(\mathbf{u})\mathbf{M}(\mathbf{x} - \mathbf{u})d\mathbf{u} = \mathbf{F}(\mathbf{x}) \otimes \mathbf{M}(\mathbf{x}) \qquad (9)$$

which represents a convolution process. We note that Eq. (9) can be efficiently calculated using Fourier-transform (FT) convolution methods as the locality introduced by the self-shadowing effect is absent, as the convolution theorem states that:

$$\mathbf{P}(\mathbf{x}) = \mathcal{F}^{-1}\{f(\mathbf{k}) \cdot m(\mathbf{k})\} \qquad (10)$$

where $\mathcal{F}^{-1}$ is the inverse Fourier transformation, $f(\mathbf{k})$ and $m(\mathbf{k})$ are the Fourier transformations of **F** and **M** on the frequency **k** domain, respectively.

Equation (8) has several geometric implications for the operational range of MBHL, which are further discussed below:

## Critical deposition angle

From the geometry shown in Supplementary Fig. 16, the aspect ratio of the nanoaperture areas in the stencil membrane limits the highest $\varphi$ angle of the incident particle. For each particle with a location of incidence $\mathbf{x}^T$ and azimuthal angle $\theta$, the critical incident angle $\varphi_c(\mathbf{x}^T)$ is calculated by:

$$\varphi_c(\mathbf{x}^T) = \text{argmax}|_\varphi R_m(\mathbf{x}^T, \theta, \varphi) \tag{11}$$

Any particle comes from the trajectory $(\theta, \varphi)$ to $\mathbf{x}^T$ cannot reach the substrate, if $\varphi > \varphi_c$, regardless of the membrane-substrate gap. For circular nanoapertures with radius $r$, the maximum $\varphi_c$ at anywhere on the stencil surface, $\hat{\varphi}_c$ is given by $\hat{\varphi}_c = \tan^{-1}\frac{2r}{\delta}$. Supplementary Fig. 17 shows the distribution of $\varphi_c$ for circular nanopore stencils with varied aspect ratios. For the smallest holes used in the study (diameter $\approx$ 100 nm) and typical SiN$_x$ membrane with $\delta$ = 50 nm, the $\hat{\varphi}_c$ = 63.4°, which is the upper bound of tilting angle when designing the MBHL pattern.

## Self-shadowing effect

The self-shadowing effect arises from the finite thickness of the stencil membrane ($\delta$), which restricts the trajectories of incident particles. The combined effects of criteria (2) and (3) in deriving Eq. (8) can be encapsulated by the shadowing factor $\varepsilon$, defined by the ratio between $R_m$ and the critical dimension of the nanoaperture, when $\mathbf{x}^T$ is at the edge of the nanoaperture. $\varepsilon$ characterizes the proportion of the shadow cast by the membrane wall on the projected pattern, as shown in Supplementary Fig. 18. A larger $\varepsilon$ value indicates greater distortion of the resulting pattern from the nanoaperture design. For circular nanoapertures with radius $r$, $\varepsilon = (\delta \tan\varphi)/(2r)$.

In this study, we design MBHL trajectories with negligible self-shadowing effects ($\varepsilon < 0.15$), particles pass freely through the apertures without significant obstruction. For the nanoapertures used in this study, with $\delta$ = 50 nm and the smallest $r$ = 50 nm, $\varepsilon$ is only 0.04 when $\varphi$ = 5°. Supplementary Fig. 19 illustrate the influence of $\varepsilon$ on deposition patterns for various aperture designs. The combined effects of the critical deposition angle and self-shadowing effect divides the parametric hemisphere $(\theta, \varphi)$ into different operational regimes, as shown in Supplementary Fig. 20.

One potential application of the self-shadowing effect is to achieve MBHL structures with critical dimensions significantly smaller than those of the nanoapertures though high-$\varphi$ depositions, as demonstrated in Supplementary Figs. 21 and 22. This possibility will be explored in future studies.

## Deposition thickness of MBHL

Theoretically, the thickness of the deposited structure $\mathbf{H}(\mathbf{x})$, a 2D function representing the thickness at each point $\mathbf{x}$, depends on the probability $\mathbf{P}(\mathbf{x})$. Consider a time-dependent flux function $\Phi_0(t)$, as measured by the nominal thickness of deposited materials per unit time, a corresponding time-dependent $\mathbf{H}(\mathbf{x}, t)$ can be calculated as:

$$\mathbf{H}(\mathbf{x}, t) = \int_0^t \Phi_0(\tau)\mathbf{S}(\mathbf{x}, \tau)\mathbf{P}(\mathbf{x}, \tau | \mathbf{M}(\mathbf{x}, \tau), \delta(\tau), R_s(\tau))\,d\tau \tag{12}$$

where $\mathbf{S}$ is a transfer function representing the change of height due to effects like surface segregation, island merging, etc. The nanoaperture pattern ($\mathbf{M}$), membrane thickness ($\delta$), and surface diffusion length ($R_s$) are all treated as time-dependent variables, influenced by the material deposited on the top surface of the membrane and changes in surface chemistry during the deposition process.

While detailed multiscale simulations are challenging to achieve, our first-order approximation accounts for the incremental increase in membrane thickness ($\Delta\delta$) due to material accumulation. Supplementary Fig. 23 demonstrates that even with a 100 nm increase in

membrane thickness, the resulting deposition patterns remain almost invariant compared to the initial values at the beginning of the deposition. Under such considerations, the normalized $\mathbf{H}(\mathbf{x})$ profile can still be effectively approximated by $\mathbf{P}(\mathbf{x})$.

## Numerical simulations

The numerical simulations of the MBHL deposition process were conducted using the Python package mbhl available at https://github.com/alchem0x2A/nanolitho, which is distributed under the MIT license. Generally, an MBHL experiment were described as the combination of Stencil, Trajectory and Physics, where Mask corresponds to the nanoaperture design, Trajectory refers to the $(\theta, \varphi)$ parameters and Physics describes other physical parameters (separation $D$, directional diffusion length $R_d$ and non-directional diffusion parameter $\sigma_f$). Periodic nanoaperture designs were described by both the rectangular periodic unit cell and nanoaperture sites. The typical nanoaperture designs used in this work are listed as follows:

Square:       Unit cell (L, L),       Sites: [(0, 0)]
Hexagonal:    Unit cell (L, $\sqrt{3}$L),   Sites: [(0, 0), ($\frac{1}{2}$L, $\frac{\sqrt{3}}{2}$L)]
Honeycomb:          Unit         cell        ($\sqrt{3}L$, $3L$),
Sites: [(0, 0), ($\frac{\sqrt{3}}{2}L$, $\frac{1}{2}L$), ($\frac{\sqrt{3}}{2}L$, $\frac{3}{2}L$), ($\sqrt{3}L$, $2L$)]

In these designs, the nanoapertures have uniform radii of $r$ and the closest spacing between two neighboring apertures is $L$.

The mbhl package offers simulation of the deposition using various combinations of geometry (periodic or non-periodic lattice) and simulation algorithms. 3 versions of methods can be chosen to calculate the deposition pattern:

1. The naïve direct method: Computes deposition by explicitly simulating overlap between each deposition pattern from n-beam interference, offering high accuracy but at a significant computational cost. Used as a benchmark method

2. The raytracing method: Incorporating the full treatment from Eq. (8) and is in general more than 1 order of magnitude faster than the direct method

3. The fftconvolve method: Leveraging the efficient convolution theorem (Eq. (10)) in Fourier space. This method ignores the self-shadowing effect.

A comprehensive comparison of the performance of these methods is provided in Supplementary Fig. 24. While fftconvolve is ideal for rapid calculations in systems with negligible self-shadowing (applicable for all examples involved in this study), the raytracing method is recommended for scenarios requiring high-$\varphi$ deposition, such as in Supplementary Figs. 21 and 22.

## Derivation of Eq. (2)

Supplementary Fig. 25 shows the detailed trigonometric relationships between neighboring nanopores in periodic nanoapertures. The length of an intersecting arc, $C_i$, is determined by its central angle $\alpha_i$ and radius $R$ of the offset trajectory, such that $C_i = \alpha_i R$. Due to symmetry, there are $N_i$ equivalent arcs intersecting with the $i$-th nearest nanopores. $P_c$ can thus be expressed as the ratio of the total length of all intersecting arcs to the circumference of the filter ring:

$$P_c = \begin{cases} \frac{\sum_{i=1}^\infty N_i C_i}{2\pi R} = \frac{\sum_{i=1}^\infty N_i \alpha_i}{2\pi}, & \frac{R}{r} > 1 \\ 1, & 0 < \frac{R}{r} \leq 1 \end{cases} \tag{13}$$

The value of $\alpha_i$ for an intersecting arc $C_i$ is calculated using the law of cosines for the triangle formed by $R$, $r$ and $L_i$:

$$\alpha_i = 2\cos^{-1}\left[\frac{\left(\frac{R}{r}\right)^2 + \left(\frac{L_i}{r}\right)^2 - 1}{2\frac{R}{r}\frac{L_i}{r}}\right] \tag{14}$$

When the trajectory pattern does not intersect with $i$-th NNs, there is no real solution for $\alpha_i$. We arrive at Eq. (2) by further taking into account that $P_c$ must be 1 for any $r < R$. The number of equivalent $i$-th NNs, $N_i$, is defined as the number of neighboring nanopores with the same center-to-center length $L_i$. For 2D square and hexagonal lattices, the values for $L_i$ are

$$L_i = \begin{cases} L \cdot \sqrt{h^2 + k^2}, & \text{for square lattice} \\ L \cdot \sqrt{\left(h + \frac{1}{2}k\right)^2 + \left(\frac{\sqrt{3}}{2}k\right)^2}, & \text{for hexagonal lattice} \end{cases} \quad (15)$$

where $h$ and $k$ are integers $(0, \pm 1, \pm 2, \ldots)$. The first few positive $L_i$ values for a square lattice are $\{L, \sqrt{2}L, 2L, \sqrt{5}L, 2\sqrt{2}L, \ldots\}$, while those for a hexagonal lattice become $\{L, \sqrt{3}L, 2L, \sqrt{7}L, 3L, \ldots\}$, corresponding to the straight lines with constant $R/L$ values in Fig. 3c,e.

## Dimensionality characterization of MBHL patterns

We use the algorithm presented by Larsen et al.[34]. to determine the dimensionality of the deposition pattern. Analogous to the chemical bonding, the neighboring deposited circles $i,j$ are considered "connected" if their center-to-center distance $d_{i,j}$ is smaller than the summation of radii, such that $d_{i,j} < k(r_i + r_j)$, where $k$ is the multiplier on the cutoff distance. We used a $k$-cutoff of 1.05 to separate the 0D, 1D and 2D components of the formed patterns in all the simulations.

## Explanation of Eq. (3)

Despite the existence of various mathematical measures of geometric chirality[37–39], the majority of these approaches are based on discrete point clouds (i.e. on discrete atomic positions), which is not easy to be adapted for the continuous nanosurface with varying heights in MBHL. To quantify the degree of geometric chirality of MBHL deposition patterns, we used a similar approach as the "maximal overlap" measure proposed by Schäferling[35]. For convenience we assign $P_+$ and $P_-$ as to the patterns formed by trajectories that mirror each other. In the case of $n$-beam interference, that corresponds to trajectories with angle displacements of $+\theta_0$, respectively. By applying a translation $(u, v)$ followed by rotation with angle $\tau$, the normalized height difference between $P_+$ and transformed $P_-$ on the same numerical grids are compared. For clarity, $S$ increases with higher degree of chirality, which is slightly different from the original measure proposed by Schäferling, which becomes 1 when the structure is achiral.

## Data availability

The data that support the findings of this study are available from the corresponding author upon request. Source data are provided with this paper.

## Code availability

The CL simulation code and data is available at Zenodo[40], which is distributed under the MIT license, and from the corresponding author.

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

## Acknowledgements

C.J.S. acknowledges financial support from ETH Zürich, the Swiss National Science Foundation (Project number: 223243) and the European Research Council (Grant number: ERC-stg-849229). S.S.Z. is grateful for technical support from the Binnig and Rohrer Nanotechnology Center (BRNC) at IBM research center in Zürich.

## Author contributions

S.S.Z. and C.J.S. conceived the idea and designed the experiments. S.S.Z. carried out the nanopatterning of metals and metal oxides. T.T. established the computational lithography algorithm and carried out all numerical analysis. J.O. performed the nanopatterning of organic semiconductors. S.S.Z. and J.O. carried out AFM and SEM characterization. Z.L. performed the optical simulations. S.S.Z., T.T. and C.J.S. co-wrote the manuscript. All authors discussed the results and commented on the paper.

## Competing interests

A patent application based on the MBHL technique has been submitted to the European Patent Office (application number: EP24185086.6) by S.S.Z., J.O. and C.J.S. The remaining authors declare no competing interests.
