## [Transparent Peer Review file · Nature Communications]

Direct Nanopatterning of Complex 3D Surfaces and Self-Aligned Superlattices via Molecular-Beam Holographic Lithography

Corresponding Author: Professor Chih-Jen Shih

Version 0:

Reviewer comments:

Reviewer #1

(Remarks to the Author)

I have carefully considered the manuscript "Direct Nanopatterning of Complex 3D Surfaces and Self-Aligned Superlattices via Molecular-Beam Holographic Lithography" which is interesting and extension of the work of NSL (Nanostencil Lithography) by incorporating computational lithography for various evaporating materials. This work is commendable as it uses Python and Matplotlib which strengthens the concept of applicability. The authors also described in detail the usage of various evaporating materials for justifying the concept of MBHL (Molecular-Beam Holographic Lithography). After careful evaluation of the manuscript, I recommend to consider the manuscript for major revision addressing the following comments:

1. Author used silicon nitride membrane in MBHL process, what is the effect of volatile organic materials? Is the interaction mechanism the same for metal oxides and volatile organic materials using MBHL method? What impact does this have on the properties of the resulting superlattices?
2. Can the authors provide the optimal range of values (for quality and uniformity) for the angular projections generated by the MBHL concept?
3. Can the authors elaborate why the tilting angle used is only 5° ?
4. Authors are suggested to provide the below information in Results and Discussion section.
What are the long-term stability and durability of the 3D patterns created through MBHL under various environmental conditions?
5. Authors can incorporate the information of what is the thickness of the pattern getting generated? Is the thickness of the pattern uniform throughout the layered structures? Does the deposition process effect the final morphology of the pattern?
6. Can the authors clearly specify the design parameters of nanoapertures (e.g., size, shape, spacing) most significantly influence the angular distribution of the molecular beam and its interference patterns? Also, explain performance of this computational lithography in terms of processing speed and accuracy of self-alignment in complex patterns?
7. In Page 3, authors specified " $n = \infty$, interference yields the most complex patterns, it is also the easiest to be implemented in practice". This statement looks contradicting for the observed results.
8. What would be the techniques that can be used for generating the patterns for $n = \infty$?
9. Authors should explain how the structural characteristics of these lattices influence their performance in the applications like nanoimaging, sensing, catalysis, optoelectronics? Also, describe the mechanical properties of the resulting superlattices?
10. Can the authors perform the comparison of computational and experimental interference 2D patterns for various values of parameter R/L ?
11. Can the authors check the captions for the figures? Such as Fig: 3(f)
12. Can the sentence on Page 2 line 59 be re-written? Authors created confusion using and/or combinations?

Reviewer #2

(Remarks to the Author)

Reviewer #3

(Remarks to the Author)

This paper by Zeng et al. introduced the development of Molecular-Beam Holographic Lithography (MBHL) for the fabrication of complex surface patterns. The key difference between the MBHL technique and the conventional glancing angle deposition technique is the free space between the nanoaperture membrane and the substrate, which allows for tuning the lateral offset of patterns besides adjusting the pattern morphologies. As a result, the MBHL technique enables to fabricate more complex surface patterns on the substrates which cannot be fabricated by the conventional glancing angle deposition technique. I think that there is enough in this work that it merits publication in Nature Communications. Therefore, my recommendation is to publish once the following comments have been addressed:

1. There is no discussion about the role of the thickness of the nanoaperture membrane (δ) in the MBHL technique.
2. As known, the surface diffusion behavior also depends on the substrate temperature. How about the substrate temperature during the deposition of different materials? And how does the temperature affect the surface diffusion length (R_s)?
3. In the deposition process, whether the substrate is rotating or not? If the substrate is rotating during the deposition process, what is the rotating speed and how does the rotating speed affect the morphologies of the surface patterns?
4. The optical properties of the obtained surface patterns should be presented to demonstrate the application potential of the MBHL technique.
5. "Molecular-Beam" in Molecular-Beam Holographic Lithography (MBHL) may cause ambiguity in comparison to that in Molecular Beam Epitaxy.

Reviewer #4

(Remarks to the Author)

The authors report on a novel lithography technique that they term Molecular-Beam Holographic Lithography, based on shadow deposition through a suspended mask at different angles. Although multi-angle suspended shadow mask techniques are already commonly employed in nanolithography (see, for example, Appl. Phys. Lett. 99, 102107 (2011)), the authors have pushed this technique further: the resulting projections produce complex Moiré-like interference patterns that could not otherwise be fabricated using conventional techniques. The results are really quite beautiful. Further, the authors have complimented their experimentally fabricated structures with simulated ones that are able to accurately predict the resulting interference patterns. The simulations are a useful tool for predicting the fabricated nanostructures prior to dedicating time and resources to the experimental fabrication.

The manuscript is very well written, with clear descriptions of the data and sound discussion of the underlying mathematics. I believe this work will be of great interest to researchers fabricating complex nanostructures across a host of different fields, as it demonstrates a way to unlock 3D topographies that would be otherwise difficult to achieve. Further, the technique proposed here appears to be applicable to wide range of materials, with the only requirement that they can be grown with a line-of-sight deposition technique, e.g. electron beam or thermal evaporation.

I have a few questions and comments for the authors, which I think would be of interest to a reader and useful for someone wishing to replicate this technique.

1. The authors convincingly demonstrate that diffusion of the deposited material affects the final pattern. They attribute the extent of diffusion to the particle-surface interaction, which is a material-dependent effect. However, I expect that the temperature of the evaporated material will also play a role. Material deposited at higher temperatures/energies will have higher kinetic energy and would presumably diffuse further. Can the authors comment on this? If the effect is relatively temperature independent, I expect this would be useful for the reader to know.
2. Related to the previous comment, could the authors include deposition rates in the Methods?
3. The thicknesses of deposited films does not appear to have been explicitly stated. Due to the interference patterns and 3D topography I understand this is complex, but it would be useful to know the maximum height or the equivalent thickness if deposited without the nanoaperture chip.
4. Material will be deposited on top of the nanoaperture chip, effectively increasing its thickness (δ in Fig 3b). δ is already quite thin as fabricated (50 nm), and typical deposition thickness will be comparable. This will have an impact on the deposition projection, $R_m = \delta \tan(\phi)$, particularly if the aperture size is comparable to R_m . Can the authors comment on this? What is a reasonable amount of material that can be deposited before this effect becomes substantial?
5. It's not clear to me how the nanoaperture chip is secured on the substrate chip. Does it adhere to the photoresist? How is it then removed? Do the authors also remove the photoresist? On line 154, the authors make the claim that "MBHL technique presented here represents the only approach allowing direct 3D nanopatterning of the solvent-sensitive organic semiconductors." However, necessary post-processing of the chip (e.g. removal of the photoresist in DMSO) may undermine this claim.
6. The mask designs for the "merged lattice" and "waffles" patterns in Fig 4 appear to be missing from either the main manuscript or the SI. Could the authors include these, where appropriate?
7. The two sentences starting "We fabricated a series..." from lines 167 to 169 appear to me to be out of the place. Can the authors check this?

Overall, the manuscript was an enjoyable and very interesting read.

Reviewer #5

(Remarks to the Author)

The authors demonstrated a novel direct nanopatterning method, Molecular-Beam Holographic Lithography (MBHL), for patterning complex 3D surfaces and fabricating self-aligned superlattices using molecular beams. The technique aims to overcome the limitations of existing lithography methods, such as material compatibility issues and limited pattern complexity, by employing a moiré interference pattern created by angular molecular beams. Here, the authors build a computational lithography (CL) method that plays a crucial role in the MBHL technique by guiding the design and formation of complex interference patterns on the substrate surface. Although there are limitations in demonstrating experimentally, the authors theoretically build a model that MBHL enables the fabrication of any 2D periodic design. Also, it is successfully demonstrated experimentally on a given system. MBHL is a promising and innovative approach to direct nanofabrication that could affect the fields of materials science and nanotechnology. After addressing the following comments and questions, I recommend this paper for publication.

Comments to the Authors

#1. In the fabrication process, the separation between the substrate chip surface and nanoaperture membrane, D , is a very important factor precisely defined by the PR thickness. The thickness of the spacer is controlled by two different PRs, which are set as 0.7 μm and 2.5 μm . What is the range of spacer thickness within experiments? And do they affect the performance of MBHL? Additionally, when clamping the nanoaperture chip and substrate chip, is there any gap or any issues that hamper precise control of the D ? More details on the fabrication process and the information on the distribution of D would help understand the accuracy of this method. Overall, the discussion on the precision required for the experimental setup will further strengthen the experimental feasibility of implementing the theoretical development of this study.

#2. What is the level of reproducibility and repeatability when experimentally demonstrating the deposition process? Is it possible to achieve the same deposition morphology as predicted by computational lithography using only nanoaperture design and evaporation control?

#3. The deposition possibility, $P(x)$, is the 2D convolution operation of the nanoaperture pattern function, $M(x)$, and the offset trajectory function, $F(x)$. Here, the description of $P(x)$ as the deposition probability at specific points on the substrate needs to clearly state whether this function can directly represent the height of the resulting 3D morphology. For a comprehensive understanding of the deposition process, including a detailed explanation of how $P(x)$ correlates with the actual height profile of the 3D morphology would be informative. If $P(x)$ alone cannot predict the height, please discuss any additional factors or parameters required to model the 3D morphology.

Version 1:

Reviewer comments:

Reviewer #1

(Remarks to the Author)

After thoroughly reviewing the revised manuscript and the author's responses to reviewer comments, I am pleased to recommend this manuscript for publication. The authors have addressed all my comments, thereby enhancing the clarity and impact of the work. The manuscript is now well-written and provides a comprehensive description of the data, making it suitable for publication in Nature Communications.

(Remarks on code availability)

Reviewer #2

(Remarks to the Author)

(Remarks on code availability)

Reviewer #3

(Remarks to the Author)

The authors have addressed my comments, the paper could be accepted for publication now.

(Remarks on code availability)

Reviewer #4

(Remarks to the Author)

I thank the authors for their considered and detailed response to the questions and comments. I am satisfied with their answers and the corresponding changes to the manuscript, and I recommend for publication.

(Remarks on code availability)

The code is clear with specific Jupyter notebooks to reproduce the results of the paper. The repository includes a README to assist installing and running the code and reproducing the code, and will be a useful resources for researchers in the community wishing to simulate the same or similar results.

Reviewer #5

(Remarks to the Author)

The authors have done a good job in responding to comments from the referees. I believe that the revised version is suitable for publication in its current form.

(Remarks on code availability)

Response to reviewer comments

Reviewer #1 (Remarks to the Author):

I have carefully considered the manuscript “Direct Nanopatterning of Complex 3D Surfaces and Self-Aligned Superlattices via Molecular-Beam Holographic Lithography” which is interesting and extension of the work of NSL (Nanostencil Lithography) by incorporating computational lithography for various evaporating materials. This work is commendable as it uses Python and Matplotlib which strengthens the concept of applicability. The authors also described in detail the usage of various evaporating materials for justifying the concept of MBHL(Molecular-Beam Holographic Lithography).

Reply: We appreciate the positive and constructive comments made by the Reviewer.

After careful evaluation of the manuscript, I recommend to consider the manuscript for major revision addressing the following comments:

1. Author used silicon nitride membrane in MBHL process, what is the effect of volatile organic materials? Is the interaction mechanism the same for metal oxides and volatile organic materials using MBHL method? What impact does this have on the properties of the resulting superlattices?

Reply: Silicon nitride is a very stable, robust and inert inorganic material that has been extensively used in the MEMS fabrication to obtain suspending structures such as free-standing membranes in our MBHL process. Silicon nitride reactions in the literature all involve very harsh conditions. For example, its wet etching process uses boiled phosphoric acid and does not react with other wet chemicals. The dry etching uses extremely reactive fluorine-based chemicals such as CHF_3 or CH_2F_2 in plasma enhanced conditions. We could not find any literature suggesting chemical reactions taking place between SiN_x and volatile organic materials.

For example, for the evaporation of $\text{Ir}(\text{ppy})_3$, whose phosphorescence is very sensitive to reactive species, through the SiN_x nanoapertures (Figs. 2c and 2d), we did not observe any phosphorescence quenching as compared to the deposited bulk films. Since SiN_x does not react with organic compounds nor metal oxides, the difference of their interaction with SiN_x membrane is expected to be negligible.

It is noteworthy that the evaporation of these materials in high vacuum can deposit on the surface of the silicon nitride membrane. Therefore, the materials adsorbed inside the nanoapertures could cause partial clogging of the nanoapertures and reduce the dimensions of nanoapertures, leading to less materials passing through the nanoapertures. The overall thickness of the resulting superlattices can therefore be affected. Measures can be taken to reduce the clogging effect by coating the silicon nitride membrane with self-assembled monolayers, as presented by Marius *et al* (*Nano Letters* 2002, 2, 12, 1339–1343).

2. Can the authors provide the optimal range of values (for quality and uniformity) for the angular projections generated by the MBHL concept?

Reply: We thank the Reviewer for raising the interesting question. We would like to point out there is no “optimal range” of values, as each set of process parameters could lead to different patterns. Nevertheless, we agree with the Reviewer that there is indeed an “operational” range of values for the tilting angle. If the angle is beyond this range, no pattern will form. Please find our detailed analysis as follows.

Specifically, as shown in Figure 1e, the angular projections are determined by the equation $R = R_m + R_i + R_s$, where R_m , R_i , and R_s correspond to the contributions of membrane, membrane-substrate gap, and surface diffusion, respectively. During the MBHL process, membrane-substrate gap and surface diffusion are constant. Therefore, R_i and R_s are fixed values and do not affect the quality or uniformity of deposition. However, R_m is strongly related to the aspect ratio of the nanoapertures, as shown in the illustration below. For nanoapertures with critical dimensions of L_1 and L_2 , the maximum tilting angle, beyond which the vapor beam can be completely blocked by the nanoaperture sidewall equals $\tan^{-1}\left(\frac{L_1}{\delta}\right)$ and $\tan^{-1}\left(\frac{L_2}{\delta}\right)$, respectively. In other words, **the higher aspect ratio for the nanoaperture, the smaller operational range for the tilting angle φ .**

Figure R1. Schematic illustration of the maximum tilting angle allowed in MBHL when evaporating through high-aspect ratio nanoapertures, where L is comparable with δ .

The limitation of the stage tilting angle, φ , due to the aspect ratio of the nanoapertures is similar to the self-shadowing effect in glancing angle deposition (*J. Vac. Sci. Technol. A*, 2007, 25, 1317-1335). We have added a detailed analysis of the self-shadowing effect in MBHL and its operational ranges to the revised manuscript, as elaborated in the revised *Methods* section and Supplementary Figures S16–S21 (attached below). In brief, there are two main quantities to consider when designing the trajectory patterns in MBHL:

1. **The critical tilting angle φ_c .** At each location on the membrane, particles only reach the substrate when $\varphi < \varphi_c$. For circular nanoapertures with radius r , no deposition occurs when $\varphi \geq \tan^{-1}\frac{2r}{\delta}$, where δ is the thickness of the nanoaperture membrane, as shown in Supplementary Figures S16 and S17.
2. **The shadowing factor ε .** The shadowing factor ε characterizes the impact of the membrane thickness on the geometry of the MBHL pattern. For circular nanoapertures with radius r , ε is defined as the ratio between R_m and the diameter of the nanoapertures, such that $\varepsilon = \frac{\delta \tan \varphi}{2r}$. An increase in the tilting angle φ leads to greater deviation of the MBHL pattern from the nanoaperture shape, resulting in finer structures, as shown in Supplementary Figures S18 and S19. Please find the attached figures.

To ensure optimal quality and uniformity, the optimal tilting angle for MBHL must be determined by evaluating the aspect ratios of all nanoapertures across the chip. For the nanoaperture chips used in this

study, the SiN_x window has a thickness $\delta = 50$ nm, and the smallest nanoapertures have radius $r = 50$ nm. This implies i) the critical tilting angle $\varphi_c = 63.4^\circ$, and ii) to ensure $\varepsilon < 0.15$ (details see Supplementary Figures S20 and S21) where the self-shadowing can be ignored, $\varphi < 16.7^\circ$.

Under these geometries, the MBHL trajectory falls into the following regimes (see Supplementary Figure S20):

- A. $\varphi \ll 16.7^\circ$: The self-shadowing effect is negligible, and the deposition process follows the convolution in Equation 1 of the manuscript.
- B. $16.7^\circ < \varphi < 63.4^\circ$: The self-shadowing effect causes the deposition patterns to deviate from the convolution, requiring calculation via the raytracing method described in Equation M5. Deposition within this regime can lead to structures with critical dimensions smaller than the nanoaperture size
- C. $\varphi \geq 63.4^\circ$: No deposition will occur.

All the examples in this study fall within **regime A**, ensuring that our original analysis remains valid under the revised theoretical framework.

Figure S16. Critical deposition angle φ_c for the nanoaperture membrane. Each incident location \mathbf{x}^T corresponds to a φ_c value so that particles cannot reach the substrate when $\varphi > \varphi_c$. The maximum value for the critical deposition angle, $\hat{\varphi}_c$, corresponds to the value when \mathbf{x}^T lands on the edge of the nanoaperture.

Figure S17. Distribution of critical deposition angle φ_c for the circular nanoapertures on membrane with thickness $\delta = 100$ nm, and incident azimuthal angle $\theta = 45^\circ$. **a.** radius $r = 25$ nm, **b.** radius $r = 50$ nm, **c.** radius $r = 100$ nm, **d.** radius $r = 200$ nm. Scale bars, 200 nm. No particle can reach the substrate surface beyond maximum deposition angle $\hat{\varphi}_c = \tan^{-1} 2r/\delta$.

Figure S18. The self-shadowing effect in MBHL. **a.** Side view of the shadow cast by the nanoaperture wall in high-aspect-ratio nanoapertures. The gray area represents regions where no deposition occurs due to obstruction by the membrane wall. The shadowing factor ε characterizes the proportion of the shadow cast by the membrane wall on the projected pattern. **b.** Top view illustrating the distortion of the deposition pattern caused by the self-shadowing effect. The deposition shape deviates from the ideal nanoaperture pattern due to the increasing influence of ε .

Figure S19. Self-shadowing effect of the nanoaperture membrane in numerical simulations for a honeycomb lattice membrane with $\delta = 50$ nm, nanoaperture radius $r = 100$ nm, and center-to-center spacing $L = 400$ nm under $n = 1$ beam interference ($\theta = 30^\circ$) at different incident angles. **a.** $\varphi = 5^\circ$ **b.** $\varphi = 10^\circ$ **c.** $\varphi = 15^\circ$ **d.** $\varphi = 30^\circ$ **e.** $\varphi = 45^\circ$ and **f.** $\varphi = 60^\circ$. Scale bars, 500 nm. The locations of the nanoapertures were marked in white circles. The simulations were performed using the raytracing method on periodic nanoaperture lattice. The shadowing effect becomes prominent when $\varphi > 15^\circ$.

Figure S20. Operational regimes of MBHL for a certain nanoaperture design. The parametric hemisphere is divided into 3 regimes categorized by the range of deposition angle φ : A) The self-shadowing effect is negligible, and the deposition process follows the convolution in Equation 1 of the manuscript. B) The self-shadowing effect causes the deposited patterns to deviate from the convolution, and the deposition process is governed by Equation M5 of the manuscript. C) No deposition occurs

Figure S21. Evolution of MBHL pattern as function of polar angle φ in n -beam lithography. All simulation systems are periodic square lattice nanoaperture with $r = 50$ nm, $L = 200$ nm, $\delta = 50$ nm, and a fixed $R = 200$ nm. Scale bars, 200 nm. The factor $\varepsilon = \frac{\delta \tan \varphi}{2r}$ determines the prominence of self-shadowing effect. For practical considerations, systems $\varepsilon < 0.1$ can be efficiently simulate using the `fftconvolve` method which ignores shadowing effect, while otherwise the full treatment in `raytracing` method is a must.

3. Can the authors elaborate why the tilting angle used is only 5° ?

Reply: We thank the Reviewer for raising this question. As discussed earlier, in order to neglect the self-shadowing effect, the tilting angle should be below 16.7°. In this manuscript, we actually demonstrated MBHL process in various tilting angles from 0.5°, 1°, 2°, 5°, 7°, 10° and 13°, as shown in the results of Figure S3 and Figure 5. For the results shown in Figure 4 and Figure S4, tilting angle is fixed at 5° in order to study the effects of nanoaperture designs on MBHL. This choice aligns with our analysis of the operational regimes of φ in our response to Reviewer 1's point 2, where all the MBHL examples fall within **regime A** (negligible self-shadowing effect). In conclusion, while there is no limitation in the tilting angle φ physically, we choose to use small φ angle in this study while focusing on demonstrating the vast potential of the MBHL technique.

4. Authors are suggested to provide the below information in Results and Discussion section. What are the long-term stability and durability of the 3D patterns created through MBHL under various environmental conditions?

Reply: The 3D patterns created through MBHL are overall accumulations of the material atoms/molecules through a physical deposition process. Therefore, the patterns share the same properties as the material itself. The long-term stability and durability of the 3D patterns will behave the same as the materials under various environmental conditions. For instance, Figure R2 compared the SEM images of an $n = 2$ interference pattern of bullseye made by 50 nm Au. Two images were taken on Feb. 2024 (left) and Feb. 2025 (right), suggesting the nanopatterns remain unchanged one-year shelf storage (**Fig. R2**). The fabricated 3D surfaces are very stable in normal environmental conditions.

Figure R2. SEM image for an $n = 2$ interference pattern of bullseye made by 50 nm Au. Two images were taken on Feb. 2024 (left) and Feb. 2025 (right), suggesting after one-year shelf storage, the nanopatterns remain stable.

5. Authors can incorporate the information of what is the thickness of the pattern getting generated? Is the thickness of the pattern uniform throughout the layered structures? Does the deposition process effect the final morphology of the pattern?

Reply: The thickness information of the generated patterns is measured from AFM images, as shown in the z-axis of images in Figure 2(c and d), Figure 5(a and c) and Figure S9a. As can be seen from these images, the thickness is uniform for the same structure geometry in the 3D patterns. For instance, the tip heights of the cone structures in Figure 5a are the same, reflected from the same color in the images. The deposition process can strongly affect the final morphology of the patterns. Figure S3 shows that while varying the tilting angle, the generated patterns change dramatically.

The uniform thickness of the MBHL patterns also suggests that the deposition probability \mathbf{P} remains consistent throughout the process, which is confirmed by our numerical simulations. In brief, the calculated matrix \mathbf{P} represents the probability of deposition at any given moment during the deposition, while the actual thickness \mathbf{H} is a cumulative result of \mathbf{P} over time (more details see Methods section). When the material flux from the source, Φ_0 , remains constant, \mathbf{P} is mainly influenced by changes in the nanoaperture aspect ratio caused by deposition on the nanoaperture edges. As discussed in our responses to points 2 and 3, the self-shadowing effect is determined by the factor $\varepsilon = \frac{\delta \tan \varphi}{2r}$. For practical MBHL operations ($\delta = 50$ nm and $\varphi = 5^\circ$), even using the smallest nanoapertures ($r = 50$ nm), accumulating a layer of 100 nm of materials on the membrane increases ε to 0.13 (see simulations in Supplementary Figure S23), which remains within operational regime A (negligible self-shadowing effect), as defined in our response to Reviewer 1's point 2 (as shown in Supplementary Figure S20). This indicates that the pattern \mathbf{P} at the end of deposition is almost identical to the initial values. As a result, our simulated pattern \mathbf{P} can be effectively used a normalized height map to represent the actual height of the structure, \mathbf{H} , as verified by various AFM measurements throughout the study.

6. Can the authors clearly specify the design parameters of nanoapertures (e.g., size, shape, spacing) most significantly influence the angular distribution of the molecular beam and its interference patterns? Also, explain performance of this computational lithography in terms of processing speed and accuracy of self-alignment in complex patterns?

Reply: Among the design parameters of nanoapertures (e.g., size, shape, spacing), the size influences most significantly the angular distribution of the molecular beam and its interference patterns, under the same deposition conditions. As shown in Figure R1, nanoaperture size can affect from which angle the molecular beam can pass through the nanoapertures and therefore determine the amount and angular broadening of the beam. In general, the pattern formation is a complex process, which is demonstrated in our CL theory in Equation 1, as the deposition pattern $\mathbf{P}(\mathbf{x})$ is a convolution between the nanoaperture design $\mathbf{M}(\mathbf{x})$ (size, shape, spacing, etc.) and the trajectory of deposition beam $\mathbf{F}(\mathbf{x})$, which essentially blends features of the nanoaperture with the trajectory shape.

Regarding the computational performance of the CL model, the processing time is primarily determined by the rasterization resolution of the 2D image. To improve both accuracy and efficiency, we have revised the CL code to incorporate several features, including periodic boundary conditions, the self-shadowing effect, and multiple simulation methods for user flexibility. As shown in Supplementary Figure S24, we compared the processing times for different pattern-forming algorithms: the naïve direct method, the raytracing method, and the fftconvolve method, which leverages efficient Fourier Transform convolution by ignoring the self-shadowing effect. Since the majority of the MBHL systems in this study exhibit negligible self-shadowing (see our responses to point 2 and 3), performing a single CL simulation with a mesh spacing of

1 nm using the fftconvolve method takes less than 100 ms on a single CPU core. Additionally, the processing time for this method is independent of the number of incident beams. Even when fully accounting for the self-shadowing effect, a CL simulation using the raytracing method with a 1 nm mesh spacing and $n = 1000$ beams requires approximately 1 second on the same CPU architecture. The 1 nm mesh spacing employed is sufficient to accurately resolve the self-alignment of structures.

7. In Page 3, authors specified “ $n = \infty$, interference yields the most complex patterns, it is also the easiest to be implemented in practice”. This statement looks contradicting for the observed results.

Reply: We apologize for the confusion. We have corrected the sentence as follows.

Although the $n = \infty$ case may appear theoretically complicated, it is surprisingly straightforward to implement in practice by using a rotational setup.

8. What would be the techniques that can be used for generating the patterns for $n = \infty$?

Reply: As we explained in the response to point 7, the $n = \infty$ beam interference is achieved by fixing the angular molecular beam source, while rotating the chip stack on a motorized substrate holder tilted at an angle φ . There are two parameters to define the exact configuration at specific moment, namely the stage tilting angle φ , and stage rotation angle θ around the surface normal. When the substrate holder is revolving continuously, the trajectory of the (θ, φ) on a parametric hemisphere is equivalent to placing *infinite* evaporation sources along such trajectory, as shown in Figure 1e.

We have expanded the explanation on page 5:

Specifically, because the separation D remains constant throughout the process, the movement of continuously revolving chip stack, described by two parameters (θ, φ) , is equivalent to having an infinite number of evaporation sources placed along a continuous trajectory on the hemisphere with radius $D + \delta$. From the perspective of the chip stack, the collective incidence of angular material fluxes coming from all directions (θ, φ) reaching a point at the nanoaperture can be represented as a continuous trajectory on the surface of hemisphere of radius $D + \delta$. The actual offset trajectory writing on the substrate surface, which is a circle of variable radius $R \approx (D + \delta) \tan \varphi$, corresponds to the orthogonal projection to a tangent plane (x, y) placed at the celestial pole.

9. Authors should explain how the structural characteristics of these lattices influence their performance in the applications like nanoimaging, sensing, catalysis, optoelectronics? Also, describe the mechanical properties of the resulting superlattices?

Reply: We would like to point out that the focus of this manuscript is to present a new methodology (MBHL) for the fabrication of complex patterns of 3D surfaces and superlattices made by any evaporable materials, without using solvent-involved photolithography. The performance of the fabricated nanostructures in nanoimaging, sensing, catalysis, and optoelectronics, including their mechanical properties are rather beyond the scope of this manuscript.

In order to address the Reviewer’s question, we demonstrate the performance of our deposited patterns for applications in nanophotonics by carrying out the finite-difference time-domain (FDTD) simulations. A commercial software (Ansys Lumerical 2023) has been used to calculate the optical response of our 2D interference patterns. We rendered the detailed AFM 3D topography for the merged lattices ($R/r = 3.8$; $L/r = 4.0$) and waffle structures ($R/r = 3.8$; $L/r = 7.0$) in Figure 4(c-d) in the main text, followed by using the topography for the calculation of their reflectance spectra. As shown in **Figure R3**, the reflectance spectrum shows the narrow band grating peaks, which can be assigned to different grating mode numbers.

In other words, one can design the structure parameters, so that the reflectance spectra and peaks can be used for different applications. For example, since the spectral position of these gratings is influenced by the surrounding environment, they are often used as sensors at environmental conditions. The simulation results have now been added in the supporting information as **Figure S7**.

Figure R3. The reflectance spectrum (R) and diffraction orders for the structure of the merged lattice ($R/r = 3.8$; $L/r = 4.0$) and waffle ($R/r = 3.8$; $L/r = 7.0$) structures.

Regarding the mechanical properties of the resulting superlattices, we foresee no direct applications owing to the thin-film nature of the patterns, typically on the order of several tens of nanometers. Besides, the materials we tested are mainly metals and metal oxides, which tend to be brittle. By increasing the nanoaperture size and material thickness as well as exploring elastic materials and substrates, we may find useful mechanical properties of the deposited patterns (see *Nature Reviews Materials*, 2022, 7, 683–701).

10. Can the authors perform the comparison of computational and experimental interference 2D patterns for various values of parameter R/L ?

Reply: We thank the Reviewer for pointing this out. We have now updated the panels in Figure 4 by replacing the r and L values to dimensionless process parameters R/r and L/r to make the discussions in Figure 3(c-e) consistent. Specifically, R/r and L/r correspond to the deposition and the nanoaperture design parameters, respectively.

11. Can the authors check the captions for the figures? Such as Fig: 3(f)

Reply: We thank the Reviewer for pointing this out. We have updated the captions for Figure (c-f) accordingly.

12. Can the sentence on Page 2 line 59 be re-written? Authors created confusion using and/or combinations?

Reply: We have changed this sentence to: “On the other hand, the nanotransfer printing and charged aerosol jet techniques demand a restricted range of processing conditions, **and as a result can only operate on a relatively limited selection of materials.**”

Reviewer #2 (Remarks to the Author):

We thank the Reviewer's comments.

Reviewer #3 (Remarks to the Author):

This paper by Zeng et al. introduced the development of Molecular-Beam Holographic Lithography (MBHL) for the fabrication of complex surface patterns. The key difference between the MBHL technique and the conventional glancing angle deposition technique is the free space between the nanoaperture membrane and the substrate, which allows for tuning the lateral offset of patterns besides adjusting the pattern morphologies. As a result, the MBHL technique enables to fabricate more complex surface patterns on the substrates which cannot be fabricated by the conventional glancing angle deposition technique. I think that there is enough in this work that it merits publication in Nature Communications.

Reply: We appreciate the positive and constructive comments made by the Reviewer.

Therefore, my recommendation is to publish once the following comments have been addressed:

1. There is no discussion about the role of the thickness of the nanoaperture membrane (δ) in the MBHL technique.

Reply: We thank the Reviewer for the insightful question. Indeed, the thickness of the membrane δ will cause a drift of $R_m = \delta \tan\varphi$, as shown in Figure 1b inset. Although we have already included the realistic δ in all of our simulations using physical dimensions in all simulations in the previous version of manuscript, upon the Reviewer's suggestion, we have carried out new theoretical analysis about the role of nanoaperture membrane thickness. The new analysis is now implemented in the new Methods section (highlighted in red) . Please find the following analysis.

Derivation of Eq. (1)

Based on the general geometry of the MBHL system, as shown in Supplementary Figs. S14 and S15, for an incoming particle along the continuous trajectory \mathbf{F} to reach a point $\mathbf{x} = (x, y)$ on the substrate surface, the following preconditions must be fulfilled:

1. The incidence location arriving the top surface plane of nanoaperture membrane, $\mathbf{x}^T = (x_m^T, y_m^T)$ must be within regime Ω , namely $\mathbf{M}(x_m^T, y_m^T) = 1$.
2. When an incident particle leaves the lower surface plane of the membrane, its exiting location $\mathbf{x}^B = (x_m^B, y_m^B)$ must be within regime Ω , $\mathbf{M}(x_m^B, y_m^B) = 1$.
3. The existing location \mathbf{x}^B must be from the same aperture region as \mathbf{x}^T .
4. After the particle arrives at the substrate surface at $\mathbf{x}^S = (x^S, y^S)$, the surface diffusion results in a drifting distance R_s along the vector $\mathbf{u} = (x, y) - (x_m, y_m)$.

Note that the conditions 2 and 3 involves the “self-shadowing” effect resulting from the finite thickness of the nanoaperture membrane, where the incident particle is no longer able to pass through the membrane beyond a certain ψ angle. According to the geometry shown in Fig. S14b, the location of \mathbf{x}^B is essentially $\mathbf{x}^B = \mathbf{x} - (R_s + R_i)\hat{\mathbf{u}} = \mathbf{x} - (1 - \lambda)\mathbf{u}$, where $\hat{\mathbf{u}}$ is the unit vector of \mathbf{u} , and $\lambda = \frac{R_m}{R} = \frac{\delta \tan \varphi}{(D + \delta) \tan \varphi + R_s}$ is the correction factor considering the finite thickness of the membrane wall. To fulfill condition 3, we can therefore segment the nanoaperture function \mathbf{M} into a 2D label function \mathcal{L} to count individual holes, such that

$$\mathcal{L}(\mathbf{x}) = \begin{cases} 1, 2, \dots, k \dots, K, & \text{if } \mathbf{x} \text{ is within } k\text{-th pore} \\ 0, & \text{if } \mathbf{x} \text{ is outside } \Omega \end{cases} \quad (\text{M4})$$

To ensure that \mathbf{x}^T and \mathbf{x}^B are within the same pore region, their corresponding labels, l^T and l^B must be identical. Combining all above conditions, for a uniform material flux Φ_0 from the source, the material flux arriving at point \mathbf{x} , $\Phi(\mathbf{x})$, is given by:

$$\Phi(\mathbf{x}) = \iint_{-\infty}^{\infty} \Phi_0 w(\mathbf{u}) \mathbf{F}(\mathbf{u}) \mathbf{M}(\mathbf{x} - \mathbf{u}) \mathbf{M}(\mathbf{x} - (1 - \lambda)\mathbf{u}) \Delta_{l^T, l^B} du \quad (\text{M5})$$

where $w(\mathbf{u})$ is the weighting factor for each deposition event associated with vector \mathbf{u} , and Δ_{l^T, l^B} is the Kronecker delta of labels l^T and l^B . While Eq. M5 describes all the factors leading to the deposition pattern in MBHL, in most $n = \infty$ deposition cases, it can be simplified due to (i) the membrane thickness is negligible compared to the membrane-substrate gap, $\frac{\delta}{D} \ll 1$, and (ii) the stage rotation speed is uniform, $w(\mathbf{u}) = 1$. Most notably, the self-shadowing effect becomes negligible when λ approaches 0. Since \mathbf{M} is a binary function $\lim_{\lambda \rightarrow 0} \mathbf{M}(\mathbf{x} - \mathbf{u}) \mathbf{M}(\mathbf{x} - (1 - \lambda)\mathbf{u}) = \mathbf{M}^2(\mathbf{x} - \mathbf{u}) = \mathbf{M}(\mathbf{x} - \mathbf{u})$. Accordingly, the deposition probability $\mathbf{P}(\mathbf{x}) = \Phi(\mathbf{x})/\Phi_0$ follows:

$$\mathbf{P}(\mathbf{x}) = \iint_{-\infty}^{\infty} \mathbf{F}(\mathbf{u}) \mathbf{M}(\mathbf{x} - \mathbf{u}) d\mathbf{u} = \mathbf{F}(\mathbf{x}) \otimes \mathbf{M}(\mathbf{x}) \quad (\text{M6})$$

which represents a convolution process. We note that Eq. M6 can be efficiently calculated using Fourier-transform (FT) convolution methods as the locality introduced by the self-shadowing effect is absent, as the convolution theorem states that:

$$\mathbf{P}(\mathbf{x}) = \mathcal{F}^{-1}\{f(\mathbf{k}) \cdot m(\mathbf{k})\} \quad (\text{M7})$$

where \mathcal{F}^{-1} is the inverse Fourier transformation, $f(\mathbf{k})$ and $m(\mathbf{k})$ are the Fourier transformations of \mathbf{F} and \mathbf{M} on the frequency \mathbf{k} domain, respectively.

Equation M5 reveals a number of geometric implications within the operational range of MBHL as follows:

1. Critical deposition angle

According to the geometry shown in Supplementary Fig. S16, the aspect ratio of the nanoaperture areas in the nanoaperture membrane limits the highest φ angle of the incident particle. For each particle with a location of incidence \mathbf{x}^T and azimuthal angle θ , the critical incident angle $\varphi_c(\mathbf{x}^T)$ is given by:

$$\varphi_c(\mathbf{x}^T) = \operatorname{argmax}_{\varphi} R_m(\mathbf{x}^T, \theta, \varphi) \quad (\text{M8})$$

Specifically, any incident particle coming along the trajectory (θ, φ) to \mathbf{x}^T cannot reach the substrate if $\varphi > \varphi_c$, regardless of the membrane-substrate gap. For circular nanoapertures with radius r , the maximum φ_c at anywhere on the membrane surface, $\hat{\varphi}_c$ is given by $\hat{\varphi}_c = \tan^{-1} \frac{2r}{\delta}$. Supplementary Figure S17 shows the distribution of φ_c for circular nanoapertures with varied aspect ratios. For the smallest holes used in the study (diameter ~ 100 nm) and typical SiN_x membrane with $\delta = 50$ nm, the $\hat{\varphi}_c = 63.4^\circ$, which is the upper bound of tilting angle when designing the MBHL pattern.

2. Self-shadowing effect

The self-shadowing effect arises from the finite thickness of the nanoaperture membrane (δ), which restricts the trajectories of incident particles. The combined effects of criteria (2) and (3) in deriving Eq. M5 can be encapsulated by the shadowing factor ε , defined by the ratio between R_m and the critical dimension of the nanoaperture, when \mathbf{x}^T is at the edge of the nanoaperture. ε characterizes the proportion of the shadow cast by the membrane wall on the projected pattern, as shown in Supplementary Fig. S18. A larger ε value indicates greater distortion of the resulting pattern from the nanoaperture design. For circular nanoapertures with radius r , $\varepsilon = (\delta \tan \varphi) / (2r)$.

In this study, we design MBHL trajectories with negligible self-shadowing effects ($\varepsilon < 0.15$), particles pass freely through the apertures without significant obstruction. For the nanoapertures used in this study, with $\delta = 50$ nm and the smallest $r = 50$ nm, ε is only 0.04 when $\varphi = 5^\circ$. Supplementary Fig. S19 illustrates the influence of ε on deposition patterns for various aperture designs. The combined effects of the critical deposition angle and self-shadowing effect divides the parametric hemisphere (θ, φ) into different operational regimes, as shown in Supplementary Fig. S20.

One potential application of the self-shadowing effect is to achieve MBHL structures with critical dimensions significantly smaller than those of the nanoapertures though high- φ depositions, as demonstrated in Supplementary Figs. S21 and S22. This possibility will be explored in future studies.

Specifically, in brief, two factors related to δ determine the operational range of MBHL:

1. **The critical tilting angle** $\varphi_c = \tan^{-1} \frac{2r}{\delta}$. At each location on the membrane, particles only reach the substrate when $\varphi < \varphi_c$.
2. **The shadowing factor** $\varepsilon = \frac{\delta \tan \varphi}{2r}$. A larger ε value results in increased distortion of the deposited pattern due to the self-shadowing effect, which becomes significant when ε exceeds approximately 0.15.

The operational range of MBHL, as well as the accuracy of the CL model, depends on both φ_c and ε . Under our common experimental conditions ($\delta = 50$ nm, $D = 2.5$ μm , $\varphi = 5^\circ$),

- 1) The ratio between $R_m : R$ is only about 0.02, indicating that for conventional designs, the thickness δ can generally be neglected, and pattern formation can be determined primarily by the nanoaperture chip-substrate gap D .
- 2) The shadowing factor ε does not exceed 0.04 (even for the smallest nanoapertures with $r = 50$ nm are used), confirming that the self-shadowing effect is negligible in the examples presented in this study.
- 3) The maximum critical tilting angle for the examples in our study is 63.4° (calculated for $r=50$ nm), which is significantly larger than the tilting angle used.

To reflect this discussion, we have updated the main text and Figure 1e to include a more precise expression: $R = (D + \delta) \tan \varphi + R_s$, added detailed discussions on the critical tilting angle and self-shadowing effect in the Methods section, and incorporated supplementary analyses in Figures S16–S21, which are also attached below.

Figure S16. Critical deposition angle φ_c for the nanoaperture membrane. Each incident location \mathbf{x}^T corresponds to a φ_c value so that particles cannot reach the substrate when $\varphi > \varphi_c$. The maximum value for the critical deposition angle, $\hat{\varphi}_c$, corresponds to the value when \mathbf{x}^T lands on the edge of the nanoaperture.

Figure S17. Distribution of critical deposition angle φ_c for the circular nanoapertures on membrane with thickness $\delta = 100$ nm, and incident azimuthal angle $\theta = 45^\circ$. **a.** radius $r = 25$ nm, **b.** radius $r = 50$ nm, **c.** radius $r = 100$ nm, **d.** radius $r = 200$ nm. Scale bars, 200 nm. No particle can reach the substrate surface beyond maximum deposition angle $\hat{\varphi}_c = \tan^{-1} 2r/\delta$.

Figure S18. The self-shadowing effect in MBHL. **a.** Side view of the shadow cast by the nanoaperture wall in high-aspect-ratio nanoapertures. The gray area represents regions where no deposition occurs due to obstruction by the membrane wall. The shadowing factor ε characterizes the proportion of the shadow cast by the membrane wall on the projected pattern. **b.** Top view illustrating the distortion of the deposition pattern caused by the self-shadowing effect. The deposition shape deviates from the ideal nanoaperture pattern due to the increasing influence of ε .

Figure S19. Self-shadowing effect of the nanoaperture membrane in numerical simulations for a honeycomb lattice membrane with $\delta = 50$ nm, nanoaperture radius $r = 100$ nm, and center-to-center spacing $L = 400$ nm under $n = 1$ beam interference ($\theta = 30^\circ$) at different incident angles. **a.** $\varphi = 5^\circ$ **b.** $\varphi = 10^\circ$ **c.** $\varphi = 15^\circ$ **d.** $\varphi = 30^\circ$ **e.** $\varphi = 45^\circ$ and **f.** $\varphi = 60^\circ$. Scale bars, 500 nm. The locations of the nanoapertures were marked in white circles. The simulations were performed using the raytracing method on periodic nanoaperture lattice. The shadowing effect becomes prominent when $\varphi > 15^\circ$.

Figure S20. Operational regimes of MBHL for a certain nanoaperture design. The parametric hemisphere is divided into 3 regimes categorized by the range of deposition angle φ : A) The self-shadowing effect is negligible, and the deposition process follows the convolution in Equation 1 of the manuscript. B) The self-shadowing effect causes the deposited patterns to deviate from the convolution, and the deposition process is governed by Equation M5 of the manuscript. C) No deposition occurs

Figure S21. Evolution of MBHL pattern as function of polar angle φ in n -beam lithography. All simulation systems are periodic square lattice nanoaperture with $r = 50$ nm, $L = 200$ nm, $\delta = 50$ nm, and a fixed $R = 200$ nm. Scale bars, 200 nm. The factor $\varepsilon = \frac{\delta \tan \varphi}{2r}$ determines the prominence of self-shadowing effect. For practical considerations, systems $\varepsilon < 0.1$ can be efficiently simulate using the `fftconvolve` method which ignores shadowing effect, while otherwise the full treatment in `raytracing` method is a must.

Figure S22. Finer structures formed under high- ϕ deposition conditions leveraging the self-shadowing effect. Examples were generated using the same nanoaperture chips in Figure S18 under $n = \infty$ beam interference with: **a.** $\phi = 5^\circ, D = 2.24\mu\text{m}$. **b.** $\phi = 45^\circ, D = 150\text{ nm}$. **c.** $\phi = 60^\circ, D = 65.5\text{ nm}$. Scale bars, 200 nm. The locations of the nanoapertures are labeled in white circles. In all simulations, the radius of offset trajectory was constant at $R = 200\text{ nm}$. As ϕ increases, the self-shadowing effect intensifies, and the critical dimension of the MBHL patterns, measured using the most prominent surface features, decreases significantly, from 64 nm at $\phi = 5^\circ$ to only 13 nm at $\phi = 60^\circ$, well below the dimensions of the nanoapertures used. Notably, the small spacing ($D = 65.5\text{ nm}$) poses technical challenges that will be addressed in future studies.

2. As known, the surface diffusion behavior also depends on the substrate temperature. How about the substrate temperature during the deposition of different materials? And how does the temperature affect the surface diffusion length (R_s)?

Reply: We thank the Reviewer for raising the question. We have tested the gold deposition under various substrate temperatures from room temperature (RT), 100 °C, and 200 °C. The nanoaperture diameter is 300 nm. As we can observe in **Figure R4**, the ring width extracted by fitting the ring pattern with two concentric circles changes from 628 nm at RT, 487 nm at 100 °C to 371 nm at 200 °C, which is getting closer to 300nm as the temperature increases. Overall, the ring width decreases as increasing the temperature.

This can be explained as follows: as the temperature increases, the surface diffusion rate increases and the evaporants landing on the substrate surface move faster. If we recall how thin film is formed in a PVD process, we know that atoms tend to form islands by merging with pre-landed atoms, instead of randomly moving around, since the surface energy can be lowered. As a result, the pattern broadening effect is less profound and the grain size is larger.

On the other hand, by fitting the central radius which is identical to the offset value R , of the ring patterns (indicated by the red dashed line) using the same method as Figure S5 reveals that R also slightly increases with temperature: from 1095 nm at RT to 1150 nm at 100°C and 1344 nm at 200 °C. This suggests a larger directional drift (R_s) with increasing the substrate temperature, consistent with the fact that particles with higher kinetic energy can travel longer distance before being trapped by surface sites.

In summary, by increasing the substrate temperature, which increases the kinetic energy of the particle on the substrate surface, the offset value will increase as expected, but the pattern broadening effect is reduced. The competing and synergetic effects of pattern broadening, directional diffusion, and grain formation would make it challenging to predict the interference patterns at elevated temperatures. We would like to

point out that all computational lithography-assisted nanopatterns presented in this study were carried out at room temperature. Further investigation is required to reveal the complete picture for the prediction of pattern formation at various temperatures.

Figure R4. Ring patterns generated under various substrate temperatures.

3. In the deposition process, whether the substrate is rotating or not? If the substrate is rotating during the deposition process, what is the rotating speed and how does the rotating speed affect the morphologies of the surface patterns?

Reply: Indeed, in the deposition process of 2D interference patterns, the substrate rotation is necessary to realize $n = \infty$ interference. The rotating speed, in principle, does not influence the morphology as long as the material condensation time scale on the substrate surface, which is on the order of 10^{-3} seconds (Y.B. Zudin, “Non-equilibrium Evaporation and Condensation Processes” Springer 2019), is smaller than inverse of rotating speed. In other words, the rotating speed would require to be as high as 1,000 rpm to affect the surface morphology. In all our experiments considered in this study, the rotating speed is lower than 10 rpm. We are therefore confident that the rotating speed effects are negligible.

On the other hand, the rotating speed can be a control process parameter. For example, if the whole process is completed within the period of one substrate rotation, the rotating speed will decide how much extent the substrate rotates, as shown in the case of Figure 5d, where the rotating speed was 0.2 rpm, the substrate rotated from 0 to 270° and open-ring shape was obtained.

But for the case in Figure 4, the deposition thickness was 100 nm and deposition rate was 0.5 nm/s. So the total deposition time was 200s. During the whole deposition, the substrate rotated 16.7 revolutions in total while the rotating speed was set to 5 rpm. In each rotation, the deposited thickness is simply proportional

to the materials accumulated on the surface, without altering the overall pattern. Therefore, the rotating speed plays a nearly negligible role. On the other hand, if the deposition is terminated within a few rotations, we expect to see a degree of non-uniformity in the final patterns.

In summary, for $n = \infty$ interference patterns, we advise to tune the rotating speed to fabricate the interference patterns that accumulate more than 10 rotations of material deposits.

4. The optical properties of the obtained surface patterns should be presented to demonstrate the application potential of the MBHL technique.

Reply: We would like to point out that the focus of this manuscript is to present a new methodology (MBHL) for the fabrication of complex patterns of 3D surfaces and superlattices made by any evaporable materials, without using solvent-involved photolithography. The performance of the fabricated nanostructures in nanoimaging, sensing, catalysis, and optoelectronics are rather beyond the scope of this manuscript.

In order to address the Reviewer’s question, we demonstrate the performance of our deposited patterns for applications in nanophotonics by carrying out the finite-difference time-domain (FDTD) simulations. A commercial software (Ansys Lumerical 2023) has been used to calculate the optical response of our 2D interference patterns. We rendered the detailed AFM 3D topography for the merged lattices ($R/r = 3.8$; $L/r = 4.0$) and waffle structures ($R/r = 3.8$; $L/r = 7.0$) in Figure 4(c-d) in the main text, followed by using the topography for the calculation of their reflectance spectra. As shown in **Figure R5**, the reflectance spectrum shows the narrow band grating peaks, which can be assigned to different grating mode numbers.

In other words, one can design the structure parameters, so that the reflectance spectra and peaks can be used for different applications. For example, since the spectral position of these gratings is influenced by the surrounding environment, they are often used as sensors at environmental conditions. The simulation results have now been added in the supporting information as **Figure S7**.

Figure R5. The reflectance spectrum (R) and diffraction orders for the structure of the merged lattice ($R/r = 3.8$; $L/r = 4.0$) and waffle ($R/r = 3.8$; $L/r = 7.0$) structures.

5. “Molecular-Beam” in Molecular-Beam Holographic Lithography (MBHL) may cause ambiguity in comparison to that in Molecular Beam Epitaxy.

Reply: We thank the Reviewer for the suggestion. In fact, the “molecular beam” used in our MBHL is essentially identical to the “molecular beam” used in molecular beam epitaxy (MBE). In MBE, the solid sources are heated in quasi-Knudsen effusion cells or electron-beam evaporators until they begin to slowly sublime, followed by condensation on the target substrate. The principle is identical to the molecular beams generated by thermal or e-beam evaporation. In this regard, we consider the terminology of MBHL is appropriate.

Reviewer #4 (Remarks to the Author):

The authors report on a novel lithography technique that they term Molecular-Beam Holographic Lithography, based on shadow deposition through a suspended mask at different angles. Although multi-angle suspended shadow mask techniques are already commonly employed in nanolithography (see, for example, Appl. Phys. Lett. 99, 102107 (2011)), the authors have pushed this technique further: the resulting projections produce complex Moiré-like interference patterns that could not otherwise be fabricated using conventional techniques. The results are really quite beautiful. Further, the authors have complimented their experimentally fabricated structures with simulated ones that are able to accurately predict the resulting interference patterns. The simulations are a useful tool for predicting the fabricated nanostructures prior to dedicating time and resources to the experimental fabrication.

The manuscript is very well written, with clear descriptions of the data and sound discussion of the underlying mathematics. I believe this work will be of great interest to researchers fabricating complex nanostructures across a host of different fields, as it demonstrates a way to unlock 3D topographies that would be otherwise difficult to achieve. Further, the technique proposed here appears to be applicable to wide range of materials, with the only requirement that they can be grown with a line-of-sight deposition technique, e.g. electron beam or thermal evaporation.

Reply: We appreciate the positive and constructive comments from the Reviewer.

I have a few questions and comments for the authors, which I think would be of interest to a reader and useful for someone wishing to replicate this technique.

1. The authors convincingly demonstrate that diffusion of the deposited material affects the final pattern. They attribute the extent of diffusion to the particle-surface interaction, which is a material-dependent effect. However, I expect that the temperature of the evaporated material will also play a role. Material deposited at higher temperatures/energies will have higher kinetic energy and would presumably diffuse further. Can the authors comment on this?

If the effect is relatively temperature independent, I expect this would be useful for the reader to know.

Reply: We thank the Reviewer for the question. In a vacuum evaporator, a small change of evaporation temperature would result in a drastic increase of deposition rate. In order to precisely control the evaporation rate, the state-of-the-art thermal / e-beam evaporators directly monitor the deposition rate by a quartz crystal microbalance (QCM) sensor with a PID controller, and therefore we were not able to know or directly control the exact temperature at the evaporation source.

In order to understand the temperature effects, we have tested the gold deposition under various substrate temperatures from room temperature (RT), 100°C, and 200°C. As suggested by the Reviewer, a higher substrate temperature would also increase the kinetic energy of particle on the surface, and therefore we expect to see a larger value of offset due to the increase of surface diffusion contribution. Here the nanoaperture diameter is 300 nm. As we can observe in **Figure R6**, the ring width extracted by fitting the ring pattern with two concentric circles changes from 628nm at RT, 487nm at 100°C to 371nm at 200°C, which is getting closer to 300nm as the temperature increases. Overall, the ring width decreases with increasing the temperature.

This can be explained as follows: as the temperature increases, the surface diffusion rate increases and the evaporants landing on the substrate surface move faster. If we recall how thin film is formed in a PVD process, we know that atoms tend to form islands by merging with pre-landed atoms, instead of randomly moving around, since the surface energy can be lowered. As a result, the pattern broadening effect is less profound and the grain size is larger.

On the other hand, by fitting the central radius, which is identical to the offset value R , of the ring patterns (indicated by the red dashed line) using the same method as Figure S5 reveals that R also slightly increases with temperature: from 1095 nm at RT to 1150 nm at 100°C and 1344 nm at 200°C. This suggests a larger directional drift (R_s) with increasing the substrate temperature, consistent with the fact that particles with higher kinetic energy can travel longer distance before being trapped by surface sites.

In summary, by increasing the substrate temperature, which increases the kinetic energy of the particle on the substrate surface, the offset value will increase as expected, but the pattern broadening effect is reduced. Overall, the higher kinetic energy of the material particles results in a larger grain and narrower pattern, but with a large offset. The competing and synergetic effects of pattern broadening, directional diffusion, and grain formation would make it challenging to predict the interference patterns at elevated temperatures. We would like point out that all computational lithography-assisted nanopatterns presented in this study were carried out at room temperature. Further investigation is required to reveal the complete picture for the prediction of pattern formation at various temperatures.

Figure R6. Ring patterns generated under various substrate temperatures.

2. Related to the previous comment, could the authors include deposition rates in the Methods?

Reply: We thank the Reviewer for the suggestion. The deposition rates were 0.5 and 0.1 nm/s for metals/oxides and organic semiconductors, respectively. The information has been included in *Methods*.

3. The thicknesses of deposited films does not appear to have been explicitly stated. Due to the interference patterns and 3D topography I understand this is complex, but it would be useful to know the maximum height or the equivalent thickness if deposited without the nanoaperture chip.

Reply: The thickness information of the generated patterns have been included in most of AFM images, as shown in the z-axis values images in Figure 2(c and d), Figure 5(a and c) and Supplementary Figure S9a. As can be seen from these images, the thickness is uniform for the same structure geometry in the 3D surface patterns. For instance, the tip heights of the cone structures in Figure 5a are consistent throughout the area, as reflected from the topographic color code in the images.

4. Material will be deposited on top of the nanoaperture chip, effectively increasing its thickness (δ in Fig 3b). δ is already quite thin as fabricated (50 nm), and typical deposition thickness will be comparable. This will have an impact on the deposition projection, $R_m = \delta \tan(\varphi)$, particularly if the aperture size is comparable to R_m . Can the authors comment on this? What is a reasonable amount of material that can be deposited before this effect becomes substantial?

Reply: We thank the Reviewer for the insights. Indeed, there is an offset caused by R_m from the thickness of the nanoaperture membrane. Nevertheless, we would like to point out that in practice, R_m only accounts for a minor portion of the total drift ($R_m:R \approx \delta:D = 0.02$) since the separation D between the nanoaperture chip is significantly larger than the membrane thickness. And indeed, the thickness of the membrane δ will cause a drift of $R_m = \delta \tan\varphi$, as shown in Figure 1b inset. Upon the Reviewers' suggestions, we have carried out additional theoretical analysis about the role of nanoaperture membrane thickness. The new analysis is now implemented in the new *Methods* section (highlighted in red). Please find the following analysis.

Derivation of Eq. (1)

Based on the general geometry of the MBHL system, as shown in Supplementary Figs. S14 and S15, for an incoming particle along the continuous trajectory \mathbf{F} to reach a point $\mathbf{x} = (x, y)$ on the substrate surface, the following preconditions must be fulfilled:

5. The incidence location arriving the top surface plane of nanoaperture membrane, $\mathbf{x}^T = (x_m^T, y_m^T)$ must be within regime Ω , namely $\mathbf{M}(x_m^T, y_m^T) = 1$.
6. When an incident particle leaves the lower surface plane of the membrane, its exiting location $\mathbf{x}^B = (x_m^B, y_m^B)$ must be within regime Ω , $\mathbf{M}(x_m^B, y_m^B) = 1$.
7. The existing location \mathbf{x}^B must be from the same aperture region as \mathbf{x}^T .
8. After the particle arrives at the substrate surface at $\mathbf{x}^S = (x^S, y^S)$, the surface diffusion results in a drifting distance R_s along the vector $\mathbf{u} = (x, y) - (x_m, y_m)$.

Note that the conditions 2 and 3 involves the “self-shadowing” effect resulting from the finite thickness of the nanoaperture membrane, where the incident particle is no longer able to pass through the membrane beyond a certain φ angle. According to the geometry shown in Fig. S14b, the location of \mathbf{x}^B is essentially $\mathbf{x}^B = \mathbf{x} - (R_s + R_i)\hat{\mathbf{u}} = \mathbf{x} -$

$(1 - \lambda)\mathbf{u}$, where $\hat{\mathbf{u}}$ is the unit vector of \mathbf{u} , and $\lambda = \frac{R_m}{R} = \frac{\delta \tan \varphi}{(D + \delta) \tan \varphi + R_s}$ is the correction factor considering the finite thickness of the membrane wall. To fulfill condition 3, we can therefore segment the nanoaperture function \mathbf{M} into a 2D label function \mathcal{L} to count individual holes, such that

$$\mathcal{L}(\mathbf{x}) = \begin{cases} 1, 2, \dots, k \dots, K, & \text{if } \mathbf{x} \text{ is within } k\text{-th pore} \\ 0, & \text{if } \mathbf{x} \text{ is outside } \Omega \end{cases} \quad (\text{M4})$$

To ensure that \mathbf{x}^T and \mathbf{x}^B are within the same pore region, their corresponding labels, l^T and l^B must be identical. Combining all above conditions, for a uniform material flux Φ_0 from the source, the material flux arriving at point \mathbf{x} , $\Phi(\mathbf{x})$, is given by:

$$\Phi(\mathbf{x}) = \iint_{-\infty}^{\infty} \Phi_0 w(\mathbf{u}) \mathbf{F}(\mathbf{u}) \mathbf{M}(\mathbf{x} - \mathbf{u}) \mathbf{M}(\mathbf{x} - (1 - \lambda)\mathbf{u}) \Delta_{l^T, l^B} \mathbf{u} \quad (\text{M5})$$

where $w(\mathbf{u})$ is the weighting factor for each deposition event associated with vector \mathbf{u} , and Δ_{l^T, l^B} is the Kronecker delta of labels l^T and l^B . While Eq. M5 describes all the factors leading to the deposition pattern in MBHL, in most $n = \infty$ deposition cases, it can be simplified due to (i) the membrane thickness is negligible compared to the membrane-substrate gap, $\frac{\delta}{D} \ll 1$, and (ii) the stage rotation speed is uniform, $w(\mathbf{u}) = 1$. Most notably, the self-shadowing effect becomes negligible when λ approaches 0. Since \mathbf{M} is a binary function $\lim_{\lambda \rightarrow 0} \mathbf{M}(\mathbf{x} - \mathbf{u}) \mathbf{M}(\mathbf{x} - (1 - \lambda)\mathbf{u}) = \mathbf{M}^2(\mathbf{x} - \mathbf{u}) = \mathbf{M}(\mathbf{x} - \mathbf{u})$. Accordingly, the deposition probability $\mathbf{P}(\mathbf{x}) = \Phi(\mathbf{x})/\Phi_0$ follows:

$$\mathbf{P}(\mathbf{x}) = \iint_{-\infty}^{\infty} \mathbf{F}(\mathbf{u}) \mathbf{M}(\mathbf{x} - \mathbf{u}) \mathbf{u} \mathbf{u} = \mathbf{F}(\mathbf{x}) \otimes \mathbf{M}(\mathbf{x}) \quad (\text{M6})$$

which represents a convolution process. We note that Eq. M6 can be efficiently calculated using Fourier-transform (FT) convolution methods as the locality introduced by the self-shadowing effect is absent, as the convolution theorem states that:

$$\mathbf{P}(\mathbf{x}) = \mathcal{F}^{-1}\{f(\mathbf{k}) \cdot m(\mathbf{k})\} \quad (\text{M7})$$

where \mathcal{F}^{-1} is the inverse Fourier transformation, $f(\mathbf{k})$ and $m(\mathbf{k})$ are the Fourier transformations of \mathbf{F} and \mathbf{M} on the frequency \mathbf{k} domain, respectively.

Equation M5 reveals a number of geometric implications within the operational range of MBHL as follows:

3. Critical deposition angle

According to the geometry shown in Supplementary Fig. S16, the aspect ratio of the nanoaperture areas in the nanoaperture membrane limits the highest φ angle of the incident particle. For each particle with a location of incidence \mathbf{x}^T and azimuthal angle θ , the critical incident angle $\varphi_c(\mathbf{x}^T)$ is given by:

$$\varphi_c(\mathbf{x}^T) = \operatorname{argmax}_{|\varphi|} R_m(\mathbf{x}^T, \theta, \varphi) \quad (\text{M8})$$

Specifically, any incident particle coming along the trajectory (θ, φ) to \mathbf{x}^T cannot reach the substrate if $\varphi > \varphi_c$, regardless of the membrane-substrate gap. For circular nanoapertures with radius r , the maximum φ_c at anywhere on the membrane surface, $\hat{\varphi}_c$ is given by $\hat{\varphi}_c = \tan^{-1} \frac{2r}{\delta}$. Supplementary Figure S17 shows the distribution of φ_c for circular nanopore membranes with varied aspect ratios. For the smallest holes used in the study (diameter ~ 100 nm) and typical SiN_x membrane with $\delta = 50$ nm, the $\hat{\varphi}_c = 63.4^\circ$, which is the upper bound of tilting angle when designing the MBHL pattern.

4. Self-shadowing effect

The self-shadowing effect arises from the finite thickness of the nanoaperture membrane (δ), which restricts the trajectories of incident particles. The combined effects of criteria (2) and (3) in deriving Eq. M5 can be encapsulated by the shadowing factor ε , defined by the ratio between R_m and the critical dimension of the nanoaperture, when \mathbf{x}^T is at the edge of the nanoaperture. ε characterizes the proportion of the shadow cast by the membrane wall on the projected pattern, as shown in Supplementary Fig. S18. A larger ε value indicates greater distortion of the resulting pattern from the nanoaperture design. For circular nanoapertures with radius r , $\varepsilon = (\delta \tan \varphi) / (2r)$.

In this study, we design MBHL trajectories with negligible self-shadowing effects ($\varepsilon < 0.15$), particles pass freely through the apertures without significant obstruction. For the nanoapertures used in this study, with $\delta = 50$ nm and the smallest $r = 50$ nm, ε is only 0.04 when $\varphi = 5^\circ$. Supplementary Fig. S19 illustrates the influence of ε on deposition patterns for various aperture designs. The combined effects of the critical deposition angle and self-

shadowing effect divides the parametric hemisphere (θ, φ) into different operational regimes, as shown in Supplementary Fig. S20.

One potential application of the self-shadowing effect is to achieve MBHL structures with critical dimensions significantly smaller than those of the nanoapertures though high- φ depositions, as demonstrated in Supplementary Figs. S21 and S22. This possibility will be explored in future studies.

Specifically, in brief, two factors related to δ determine the operational range of MBHL:

3. **The critical tilting angle** $\varphi_c = \tan^{-1} \frac{2r}{\delta}$. At each location on the membrane, particles only reach the substrate when $\varphi < \varphi_c$.
4. **The shadowing factor** $\varepsilon = \frac{\delta \tan \varphi}{2r}$. A larger ε value results in increased distortion of the deposited pattern due to the self-shadowing effect, which becomes significant when ε exceeds approximately 0.15.

With the above analysis in mind, going back to the Reviewer's question, the main effect from the increase of membrane thickness is that the aspect ratio of nanoapertures shown in Figure R7 becomes increased.

Figure R7. Schematic illustration of the maximum tilting angle allowed in MBHL when evaporating through high-aspect ratio nanoapertures, where L is comparable with δ .

This change affects the critical tilting angle φ_c and the shadowing factor ε , reducing the amount of material deposited through the nanoapertures (details see revised *Methods* and Supplementary Figures S16, S18, and S23). As shown in the AFM images in Supplementary Figure S9a, the total thickness of the deposited patterns increases with the nanoaperture dimensions. Over time, as deposition continues, the membrane thickness increases, and the nanoaperture size decreases due to clogging from deposition on the sidewalls. Both factors result in reduced material reaching the substrate, leading to cone-shaped patterns in the absence of substrate tilting.

In our simulations, it is possible to account for the variation of nanoaperture aspect ratio as a function of deposition time, as detailed in the revised *Methods* section. The overall thickness \mathbf{H} can be expressed in Equation M7 as

$$\mathbf{H}(\mathbf{x}, t) = \int_0^t \Phi_0(\tau) \mathbf{S}(\mathbf{x}, \tau) \mathbf{P}(\mathbf{x}, \tau | \mathbf{M}(\mathbf{x}, \tau), \delta(\tau), R_s(\tau)) d\tau$$

where \mathbf{S} is a transfer function representing the change of height due to effects like surface segregation, island merging, etc, and the time-dependent deposition probability \mathbf{P} is influenced by the nanoaperture dimension, membrane thickness δ and surface diffusion length R_s . Although full multiscale simulations are required to fully understand the time-dependency of individual components in Equation M9, we offer a

simple insight by comparing the deposition probability matrix \mathbf{P} when increasing the membrane thickness δ due to accumulation on the top surface. As shown in Supplementary Figure S23, the values for \mathbf{P} remain nearly invariant for up to 100 nm deposition on the membrane, due to the fact that the shadowing factor ε remains small.

In conclusion, although more detailed studies are required to fully elucidate the time-dependent behavior in MBHL, our analysis shows that the deposition process aligns well with our theoretical prediction at least for a deposit of thicknesses below 100 nm.

Figure S16. Critical deposition angle φ_c for the nanoaperture membrane. Each incident location \mathbf{x}^T corresponds to a φ_c value so that particles cannot reach the substrate when $\varphi > \varphi_c$. The maximum value for the critical deposition angle, $\hat{\varphi}_c$, corresponds to the value when \mathbf{x}^T lands on the edge of the nanoaperture.

Figure S17. Distribution of critical deposition angle φ_c for the circular nanoapertures on membrane with thickness $\delta = 100$ nm, and incident azimuthal angle $\theta = 45^\circ$. **a.** radius $r = 25$ nm, **b.** radius $r = 50$ nm, **c.** radius $r = 100$ nm, **d.** radius $r = 200$ nm. Scale bars, 200 nm. No particle can reach the substrate surface beyond maximum deposition angle $\hat{\varphi}_c = \tan^{-1} 2r/\delta$.

Figure S18. The self-shadowing effect in MBHL. **a.** Side view of the shadow cast by the nanoaperture wall in high-aspect-ratio nanoapertures. The gray area represents regions where no deposition occurs due to obstruction by the membrane wall. The shadowing factor ε characterizes the proportion of the shadow cast by the membrane wall on the projected pattern. **b.** Top view illustrating the distortion of the deposition pattern caused by the self-shadowing effect. The deposition shape deviates from the ideal nanoaperture pattern due to the increasing influence of ε .

Figure S19. Self-shadowing effect of the nanoaperture membrane in numerical simulations for a honeycomb lattice membrane with $\delta = 50$ nm, nanoaperture radius $r = 100$ nm, and center-to-center spacing $L = 400$ nm under $n = 1$ beam interference ($\theta = 30^\circ$) at different incident angles. **a.** $\varphi = 5^\circ$ **b.** $\varphi = 10^\circ$ **c.** $\varphi = 15^\circ$ **d.** $\varphi = 30^\circ$ **e.** $\varphi = 45^\circ$ and **f.** $\varphi = 60^\circ$. Scale bars, 500 nm. The locations of the nanoapertures were marked in white circles. The simulations were performed using the raytracing method on periodic nanoaperture lattice. The shadowing effect becomes prominent when $\varphi > 15^\circ$.

Figure S20. Operational regimes of MBHL for a certain nanoaperture design. The parametric hemisphere is divided into 3 regimes categorized by the range of deposition angle φ : A) The self-shadowing effect is negligible, and the deposition process follows the convolution in Equation 1 of the manuscript. B) The self-shadowing effect causes the deposited patterns to deviate from the convolution, and the deposition process is governed by Equation M5 of the manuscript. C) No deposition occurs

Figure S21. Evolution of MBHL pattern as function of polar angle φ in n -beam lithography. All simulation systems are periodic square lattice nanoapertures with $r = 50$ nm, $L = 200$ nm, $\delta = 50$ nm, and a fixed $R = 200$ nm. Scale bars, 200 nm. The factor $\varepsilon = \frac{\delta \tan \varphi}{2r}$ determines the prominence of self-shadowing effect. For practical considerations, systems $\varepsilon < 0.1$ can be efficiently simulate using the `fftconvolve` method which ignores shadowing effect, while otherwise the full treatment in `raytracing` method is a must.

Figure S22. Finer structures formed under high- ϕ deposition conditions leveraging the self-shadowing effect. Examples were generated using the same nanoaperture chips in Figure S18 under $n = \infty$ beam interference with: **a.** $\phi = 5^\circ, D = 2.24\mu\text{m}$. **b.** $\phi = 45^\circ, D = 150\text{ nm}$. **c.** $\phi = 60^\circ, D = 65.5\text{ nm}$. Scale bars, 200 nm. The locations of the nanoapertures are labeled in white circles. In all simulations, the radius of offset trajectory was constant at $R = 200\text{ nm}$. As ϕ increases, the self-shadowing effect intensifies, and the critical dimension of the MBHL patterns, measured using the most prominent surface features, decreases significantly, from 64 nm at $\phi = 5^\circ$ to only 13 nm at $\phi = 60^\circ$, well below the dimensions of the nanoapertures used. Notably, the small spacing ($D = 65.5\text{ nm}$) poses technical challenges that will be addressed in future studies.

Figure S23. Simulating the effect of material accumulation on the top surface of the membrane. **a.** Schematic illustration showing the increase in membrane thickness $\Delta\delta$ due to material

accumulation on the membrane, which leads to an increase in the shadowing factor ε . **b-d.** CL simulations for the probability of deposition \mathbf{P} at different values of $\Delta\delta$ of 0 nm, 50 nm, and 100 nm, respectively. The MBHL parameters are identical to those used in Figure S19a.

5. It's not clear to me how the nanoaperture chip is secured on the substrate chip. Does it adhere to the photoresist? How is it then removed?

Do the authors also remove the photoresist? On line 154, the authors make the claim that “MBHL technique presented here represents the only approach allowing direct 3D nanopatterning of the solvent-sensitive organic semiconductors.” However, necessary post-processing of the chip (e.g. removal of the photoresist in DMSO) may undermine this claim.

Reply: The nanoaperture chip was placed on top of the substrate chip with photoresist as a spacer. The substrate chip was prepared by doing one-step photolithography to expose the silicon in the center and keep the photoresist in the surrounding area. The four corners of the nanoaperture chip were fixed by sticky Kapton tape. After the deposition, the tape was removed and the nanoaperture chip was released (see Fig. R8 below).

Figure R8. Optical images of the chips in MBHL process.

The photoresist mainly acts as a spacer between the nanoaperture chip and substrate chip to provide a defined gap D in Figure 1b. The patterns are created in the silicon area where does not expose to solvent during the patterning process. The photoresist layer does not have to remove, as the resulting nanopatterns can be directly characterized or processed for different applications. Moreover, the photoresist can also be replaced by inorganic materials such as SiO_2 or SiN_x deposited by PECVD to provide the spacer and whole substrate chip can be inert to any organic solvents.

6. The mask designs for the “merged lattice” and “waffles” patterns in Fig 4 appear to be missing from either the main manuscript or the SI. Could the authors include these, where appropriate?

Reply: Thanks for the suggestion. Figure R9, corresponding to new Supplementary Figure S9, has been added in the SI to show the mask designs for Figure 4c and 4d.

Figure R9. SEM images for the nanoaperture chip designs for the interference patterns shown in Figure 4c and 4d.

7. The two sentences starting “We fabricated a series...” from lines 167 to 169 appear to me to be out of the place. Can the authors check this?

Reply(SSZ): We checked the submitted version in the manuscript tracking system, the sentences look good, as displayed below: “We fabricated a series of 3D line surfaces made by germanium (Ge) on the (λ_1, λ_2) landscape. Our CL simulated topography nicely describes the AFM height profiles (Fig. 2f insets).”.

Overall, the manuscript was an enjoyable and very interesting read.

Reply: We thank the Reviewer again for the recognition of our work.

Reviewer #5 (Remarks to the Author):

The authors demonstrated a novel direct nanopatterning method, Molecular-Beam Holographic Lithography (MBHL), for patterning complex 3D surfaces and fabricating self-aligned superlattices using molecular beams. The technique aims to overcome the limitations of existing lithography methods, such as material compatibility issues and limited pattern complexity, by employing a moiré interference pattern created by angular molecular beams. Here, the authors build a computational lithography (CL) method that plays a crucial role in the MBHL technique by guiding the design and formation of complex interference patterns on the substrate surface. Although there are limitations in demonstrating experimentally, the authors theoretically build a model that MBHL enables the fabrication of any 2D periodic design. Also, it is successfully demonstrated experimentally on a given system. MBHL is a promising and innovative approach to direct nanofabrication that could affect the fields of materials science and nanotechnology. After addressing the following comments and questions, I recommend this paper for publication.

Reply: We appreciate the positive and constructive comments from the Reviewer.

Comments to the Authors

1. In the fabrication process, the separation between the substrate chip surface and nanoaperture membrane, D , is a very important factor precisely defined by the PR thickness. The thickness of the spacer is controlled by two different PRs, which are set as 0.7 μm and 2.5 μm . What is the range of spacer thickness within experiments? And do they affect the performance of MBHL? Additionally, when clamping the nanoaperture chip and substrate chip, is there any gap or any issues that hamper precise control of the D ? More details on the fabrication process and the information on the distribution of D would help understand the accuracy of this method. Overall, the discussion on the precision required for the experimental setup will further strengthen the experimental feasibility of implementing the theoretical development of this study.

Reply: We thank the Reviewer for the suggestion. In our experiments, we mainly tested three spacer thickness: 0 μm (*i.e.* no gap between the nanoaperture chip and substrate chip), 0.7 μm and 2.5 μm . As shown in Figure R10, the deposited ring radius, corresponding to the offset R , increases as the spacer thickness (*i.e.* separation between the nanoaperture chip and substrate chip). In theory, there is no limit of the range of spacer thickness within experiments. The spacer thickness is chosen based on how large spreading distance we want to obtain. The larger spacer/photoresist thickness, the larger spreading distance of vapor particles in MBHL.

The inset of Figure 1b shows that the offset value $R = R_m + R_i + R_s$. For example, for $D = 2.5 \mu\text{m}$ and $\varphi = 5^\circ$, $R_m = D \tan \varphi = 219 \text{ nm}$. In the nanoscale regime, this is prominent to strongly deviate the landing point of vapor particles. The photoresist thickness can be changed by varying the spin speed and resist model. Figure R11 presents the PR thinness as a function of spin rate for AZ1500 series photoresist. In our experiments, we mainly used AZ1505 and AZ1518, which can achieve a thickness range from 0.5 μm to 6 μm . For other thickness, AZ4000 series or SU-8 series can be chosen to obtain thickness from 2 μm to more than 200 μm . Besides, the spacer can also be created with inorganic materials such as SiO_2 or SiN_x deposited by means of plasma enhanced chemical vapor deposition (PECVD). The spacer height is simply controlled by the deposition thickness and meanwhile the spacer can withstand higher temperature than the photoresist during the MBHL process.

When clamping the nanoaperture chip and substrate chip, the nanoaperture chip was pressed hard and fixed tightly by Kapton tape on 4 corners to minimize additional gap apart from the photoresist thickness, as shown in the middle panel of Figure R12. Based on the “No Gap” results in Fig. R10, the effects of intrinsic gap seem negligible (the offset is very small). But to minimize the effect of the small variation of intrinsic gap from batch to batch, we suggest to fabricate the interference patterns using: (i) small tilting angles φ or (ii) large separation D , such that the offset deviation $\Delta R \approx \Delta D \tan \varphi$ is negligible. We have included the discussions in the new Supplementary information.

Figure R10. Influence of the separation between the nanoaperture chip and substrate chip on deposited ring patterns. Ge with a thickness of 100 nm was deposited at a tilting angle of 5°.

Figure R11. Photoresist thickness vs spin speed for various photoresist (Source: Microchemicals)

Figure R12. Optical images of the chips in MBHL process.

2. What is the level of reproducibility and repeatability when experimentally demonstrating the deposition process? Is it possible to achieve the same deposition morphology as predicted by computational lithography using only nanoaperture design and evaporation control?

Reply: We thank the Reviewer for raising this question. Essentially two factors influence the reproducibility of our proposed MBHL: (i) the degree of pattern lateral broadening and (ii) the deviation of offset, which takes into account the variation of gap separation, tilting angle, and lateral diffusion. We therefore examine the circle patterns that allow us to directly quantify the offset value and degree of pattern lateral broadening. We can therefore quantify the batch-to-batch reproducibility.

Specifically, we placed two pairs of nanoaperture and substrate chips at the center of substrate holder inside the evaporation system. Both nanoaperture chips have the same pattern design. We then repeated the MBHL process three times by Kapton taping regionally and selecting part of the nanoapertures on the chip open each time. The results are summarized in Figure R13. Figure R13a shows SEM images of the generated rings in different runs for each chip. Figure R13b presents the analysis of the extracted ring width and radius, which directly quantify the degree of pattern broadening and offset R , respectively, from at least 6 different rings in each SEM panel. We can see that for each run, the uniformity within the nanoaperture chip is relatively good, indicated by the small error bar. From run to run, we can see slight variation in both ring width and radius (the offset R). The run-to-run non-uniformity can be attributed to the small variation of gap distance as well as substrate surface properties surface conditions.

In practice, following our discussion in answering the Reviewer's first comment, we suggest to carry out deposition at small tilting angles φ and large separation D to mitigate the gap variation from batch to batch. In addition, a thorough cleaning process, such as piranha process or argon plasma bombardment pre-cleaning before mounting the nanoaperture chip, could also help. It is empirically useful to do a prototype test first on the substrate of interest and extract the surface diffusion factor. Then the extracted results can be used as input parameters for the computational lithography model to guide the new deposition.

Figure R13. Examination of reproducibility and repeatability demonstration with two chips for three runs of deposition.

3. The deposition possibility, $P(x)$, is the 2D convolution operation of the nanoaperture pattern function, $M(x)$, and the offset trajectory function, $F(x)$. Here, the description of $P(x)$ as the deposition probability at specific points on the substrate needs to clearly state whether this function can directly represent the height of the resulting 3D morphology. For a comprehensive understanding of the deposition process, including a detailed explanation of how $P(x)$ correlates with the actual height profile of the 3D morphology would be informative. If $P(x)$ alone cannot predict the height, please discuss any additional factors or parameters required to model the 3D morphology.

Reply: We thank the Reviewer for the insightful comment. To address this, we have updated the theoretical framework in the revised manuscript to incorporate the following effects:

The actual height profile $\mathbf{H}(\mathbf{x})$ is the cumulative result of a time-dependent probability $\mathbf{P}(\mathbf{x}, t)$ over the deposition process, and described by the following expression (Equation M9 in our manuscript):

$$\mathbf{H}(\mathbf{x}, t) = \int_0^t \Phi_0(\tau) \mathbf{S}(\mathbf{x}, \tau) \mathbf{P}(\mathbf{x}, \tau | \mathbf{M}(\mathbf{x}, \tau), \delta(\tau), R_s(\tau)) d\tau$$

Where Φ_0 is the material flux from the source, \mathbf{S} is the transfer function representing the change of height due to effects like surface segregation, island merging, *etc.* and the time-dependent \mathbf{P} matrix also relies on the nanoaperture opening (\mathbf{M}), nanoaperture membrane thickness δ , and surface diffusion R_s at each moment.

The normalized height profile $\mathbf{H}(\mathbf{x})$ can be approximated by $\mathbf{P}(\mathbf{x})$ under the following conditions:

- i. The material flux Φ_0 remains constant throughout the deposition process.
- ii. The transfer function \mathbf{S} is close to unity, meaning that the density and crystallinity of the deposited material are consistent.
- iii. The probability matrix \mathbf{P} remains identical during the deposition process.

While criteria (i) and (ii) can be managed by optimizing deposition source and experimental conditions, we have updated our theoretical framework to account for the influence of the nanoaperture aspect ratio on the MBHL deposition patterns. Details of this analysis are included in the revised Methods section. As shown in Supplementary Figure S23 (see below), even for nanoapertures with the highest aspect ratio ($\delta = 50$ nm, $r = 50$ nm), the simulated MBHL pattern \mathbf{P} remains almost unchanged after a 100 nm increase in membrane thickness due to material accumulation. This observation supports the validity of approximating the height profile $\mathbf{H}(\mathbf{x})$ using the deposition probability $\mathbf{P}(\mathbf{x})$ under the conditions presented in this study.

Figure S23. Simulating the effect of material accumulation on the top surface of the nanoaperture chip. **a.** Schematic illustration showing the increase in membrane thickness $\Delta\delta$ due to material accumulation on the membrane, which leads to an increase in the shadowing factor ϵ . **b-d.** CL

simulations for the probability of deposition \mathbf{P} at different values of $\Delta\delta$ of 0 nm, 50 nm, and 100 nm, respectively. The MBHL parameters are identical to those used in Figure S19a.